# The role of side-branching in microstructure development in laser powder-bed fusion

Minh-Son Pham [1]*, Bogdan Dovgyy[1], Paul A. Hooper[2], Christopher M. Gourlay [1] & Alessandro Piglione[1]

In-depth understanding of microstructure development is required to fabricate high quality products by additive manufacturing (for example, 3D printing). Here we report the governing role of side-branching in the microstructure development of alloys by laser powder bed fusion. We show that perturbations on the sides of cells (or dendrites) facilitate crystals to change growth direction by side-branching along orthogonal directions in response to changes in local heat flux. While the continuous epitaxial growth is responsible for slender columnar grains confined to the centreline of melt pools, side-branching frequently happening on the sides of melt pools enables crystals to follow drastic changes in thermal gradient across adjacent melt pools, resulting in substantial broadening of grains. The variation of scan pattern can interrupt the vertical columnar microstructure, but promotes both in-layer and out-of-layer side-branching, in particular resulting in the helical growth of microstructure in a chessboard strategy with 67° rotation between layers.

[1] Department of Materials, Imperial College London, South Kensington, London SW7 2AZ, UK. [2] Department of Mechanical Engineering, Imperial College London, South Kensington, London SW7 2AZ, UK. *email: son.pham@imperial.ac.uk

Additive manufacturing (AM), also as known as 3D printing, is believed to be a key enabler in the fourth industrial revolution. AM offers tremendous advantages in fabricating complex structures, freeing the designers from geometric constraints, paving the way to develop new materials whose building blocks can be carefully constructed to achieve unprecedented properties[1–3]. However, there are significant challenges in making high performance and reliable products by AM, in particular regarding metallic parts[4–7]. Such challenges inherently relate to the formation of porosity and complex microstructure development in solidification[5,6,8]. To address such challenges, in addition to many efforts in obtaining insights into processing phenomena[9–11], it is necessary to have in-depth understanding of complex microstructure development during solidification from the single-track to multi-layer depositions with various scan strategies and the influence of porosity in microstructure development in AM to increase the confidence in tailoring the microstructure, and thereby properties of alloys. It has been frequently reported that the epitaxial growth of crystals is the most dominant phenomenon governing microstructure development in the 3D printing of alloys[8], causing a columnar grain microstructure observed in almost all printed alloys such as steels, Inconel 718, Ti6Al4V, Al alloys, high entropy alloy[6,12–18]. Nevertheless, most previous studies did not show how epitaxial growth affects the morphology and spatial distribution of microstructure from single tracks to multiple tracks of deposition. While the columnar microstructure with a preferred orientation (often [001] // build direction) was frequently reported, the spatial crystallographic orientation is much more complex. For example, Piglione et al.[13] reported that there are in fact two dominantly preferred orientations (with very distinctive morphologies) that are alternating and locate at specific locations for a bi-directional scan without rotation between layers. Underlying mechanisms responsible for these morphologies, spatial distribution and crystallographic orientations are still unclear, and need to be understood as it will enable better control of microstructure to specific locations in AM.

Many studies on solidification microstructure in casting and welding have shown that the key thermal parameters such as thermal gradient and liquidus isotherm velocity govern the growth of crystals, thereby the morphologies, spatial distribution and orientations of microstructure[19–21]. In particular, Rappaz et al. carried out a fundamental study to reveal the detailed link between thermal profile, epitaxial growth and the orientation of crystals in the single deposition on the substrate of a single crystal[22]. However, AM entails many cycles of deposition mostly on polygrain substrates. Therefore, it remains important to study how the microstructure spatially develops from single tracks to multiple-track layers of deposition on polycrystal substrates. In particular, AM has a powerful capacity in changing the process parameters to effectively vary the thermal parameters of the melt pool from location to location within layers and from layer to layer, enabling the tailoring of the material microstructure to specific locations to achieve desired mechanical properties[23–25]. In addtion, it is important to highlight the influential role of scan strategy in controlling preferred texture and minimising columnar grains, residual stresses and cracking behaviour in built parts to achieve desired mechanical properties[17,24,26–28]. The opportunities of using scan strategies to tailor microstructure, thereby mechanical behaviour, reiterate the need of studying the detail of the crystal orientation, morphology, spatial distribution and length-scale of microstructure during epitaxial growth from single tracks to multi-track layers of deposition under the variation in scan strategy.

In this study, solidification microstructure (morphology, length-scale and crystallographic orientation) is examined and related to the local thermal parameters to study the underlying mechanisms responsible for microstructructure development at specific locations of melt pools in single tracks and multi-layer tracks, thereby explain the spatial distribution of microstructure and plastic anisotropy under the variation of scan strategy.

## Results

**Solidification microstructure and thermal profile**. X-ray diffraction showed that the two alloys consisted of a single face-centred cubic phase in the as-received powder and in builds fabricated by LPBF (Supplementary Fig. 1). This observation of single FCC phase was supported by electron microscopic observations by SEM and EBSD scans though nano-scale oxides were detected in transmission electron microscopy.

Figure 1a, b of a cross section transverse to a single scan track shows that the melt pool consists of multiple domains of columnar cells growing epitaxially from existing polygrains in the substrate, with their cell axis being closely perpendicular to the fusion line at locations the cells started growing from the substrate (Fig. 1b). Furthermore, cells were oriented with a <001> parallel to the cell axis (see the unit cell wireframe inset in Fig. 1a). As the local thermal gradient (G) at the fusion boundary is also normal to the boundary (Fig. 2a–c), cells have both their growth axis and a <001> orientation aligned with G (and the maximum heat flux), in good agreement with previous studies[19,20,29]. Figure 1c shows both the cross section (top and bottom regions of Fig. 1c) and longitudinal views (left and right regions of the figure) of cells in the alloy. The arithmetic average cell spacing in the printed HEA was about 0.61 μm.

Similarly, multiple domains of fine cells epitaxially growing from existing crystals formed in AM 316L (Fig. 1d). The average spacing in 316L of cells shown in Fig. 1e was 0.63 μm, almost the same as the cells in the HEA. It is frequent to see in both the alloys that cells along the centreline of melt pools of two consecutive layers kept epitaxially growing across two melt pools without changing growth direction. The axis of such cells is also closely normal to the fusion line at the bottom of the melt pool envelope, confirming the cell growth direction is anti-parallel to the thermal gradient (Fig. 2c) in agreement with previous studies[19,20]. Figure 1d, e indicate that the cells have a rod-like structure. A higher magnification image of the longitudinal view of a cell domain shows that rod-like cells have an undulated surface. Because cells can be only seen after chemical etching, the undulated surface results from the chemical perturbation. This indicates that although the high cooling rate can prohibit the formation of secondary arms, there are still solid-liquid interface instabilities in the direction orthogonal to the primary growth direction. The presence of such side instabilities indicates that cells are in the transition from cellular to dendritic growth[30–36], similar to that seen in laser and electron welding[20,37]. Because the HEA and 316L are cubic crystals, all cells of the same domains grew along a <001> orientation in agreement with previous studies[22,33,38]. To have a better picture of cell spatial development from the location from which they epitaxially grow to the top of melt pool, the local variation in the cell spacing in the cellular region shown in Fig. 1d was measured as a function of position from the bottom of melt pool. The spacing measured in this location increased from 0.43 μm near the bottom to 0.59 μm towards the top of the melt pool. The measured spacing near the bottom is in agreement with the reported spacing of primary dendrite arms seen in low Cr/Ni stainless steels fabricated by electron beam welding with a high speed of 5 m s$^{-1}$[38,39].

For a given alloy composition, the governing factors for microstructure development (including the length scale) are (i) the direction and magnitude of the thermal gradient in the

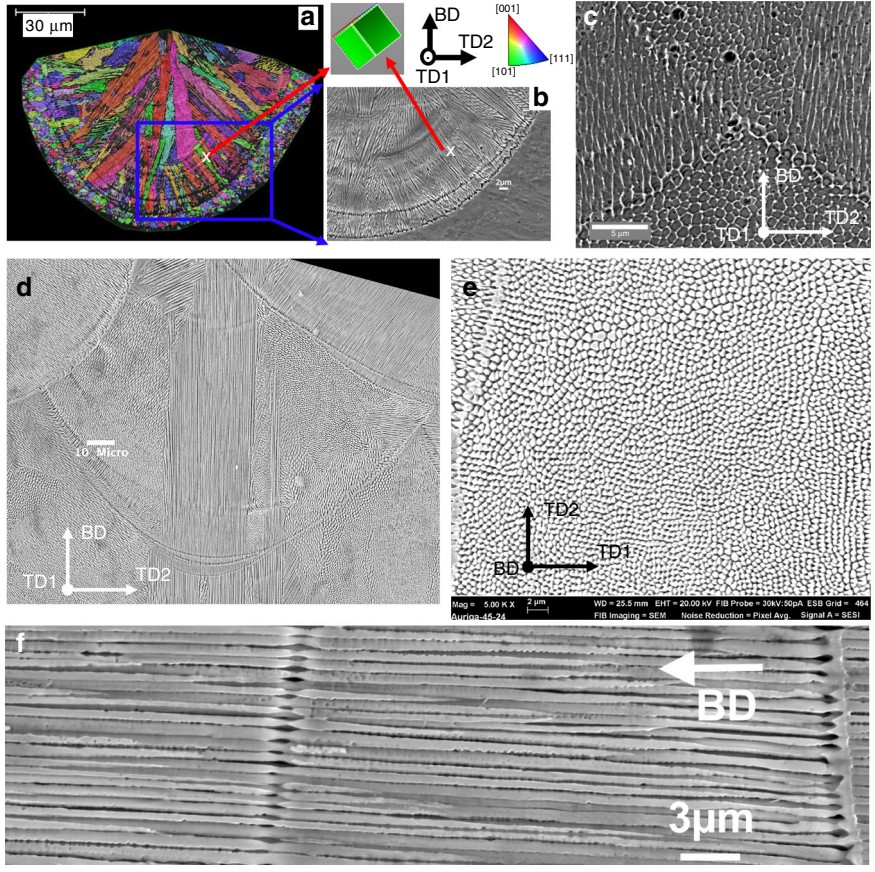

**Fig. 1 Solidification microstructure in a single-track and in a multi-layer build of HEA. a** EBSD inverse pole figure (IPF) map along build direction (BD) and **b** secondary electron image of cells in a single track of HEA. **c** Cells in regions far away from the build/substrate interface. **d–f** Solidification microstructure in 316L fabricated by a Renishaw: **d** Cells in a melt pool of 316L, **e** and **f** Transverse and longitudinal sections of a cell domain within a melt pool, respectively. Note: Figure **a** and **b** were re-used from ref. [13] under the terms of the Creative Commons Attribution License (CC BY).

liquid, **G**, and (ii) the speed of the liquidus isotherm, $v_i$. Therefore, **G** and $v_i$ were calculated along the liquidus line and within a steady state melt pool simulated by FEA (Fig. 2a–d). The FEA simulation was validated on the basis of matching the melt pool dimensions (Supplementary Note 1). In addition, the direction of the thermal gradient **G** at the fusion line is normal to the fusion line as shown in Fig. 2b, c in agreement with previous studies[20], confirming the relationship between the heat flux and the growth direction of cells. Because each melt pool was deposited vertically, the **G** direction in the centre is vertical and does not change towards the top of the melt pool. In contrast, **G** on the sides of the melt pool gradually changes direction when approaching the melt pool centre (Fig. 2c).

The spacing of cells (or primary dendrites) $\lambda_c$ can be related to $G$ and $v_i$ (which are the magnitudes of **G** and $v_i$ along the growth direction, respectively) by an equation of the form[21,40]:

$$\lambda_c = a v_i^{-m} G^{-n} \qquad (1)$$

where $m$ and $n$ are material constants, and $a$ is an alloy-dependent factor.

Past work has often assumed $m = n$ to estimate the influence of cooling rate ($dT/dt = (dT/dx)(dx/dt)$, i.e. $G \times v_i$) on the spacing of cells (or primary dendrites), for example $m = n \approx 0.5$[41,42] or $m = n \approx 0.33$[38,39,43,44]. Consequently, $(v_i^{-0.5}G^{-0.5})$ and $(v_i^{-0.33}G^{-0.33})$ were used to predict the spacing of microstructure in additively manufactured alloys. Figure 2e shows that although they both can be used to predict the spacing at a given condition, the latter appears to better predict the increasing trend in spacing.

While the product $(v_i^{-m}G^{-n})$ governs the scale of microstructure, $(G/v_i)$ controls the solidification mode (such as planar, columnar or equiaxed)[19,21,25], which in turn also affects the scale of the microstructure. Even without changing print parameters, FEA simulation shows that $(G/v_i)$ substantially increases by several orders of magnitude along the fusion line from the bottom to the top of the melt pool (Supplementary Fig. 3d – solid line). Because $v_i = v_b cos\theta$ with $\theta$ theoretically being 90° at the bottom of melt pools, $v_i$ goes to zero and $(G/v_i)$ becomes infinitely high at the very bottom of the melt pool (Supplementary Fig. 3). Although Chen et al. argued that there would be no planar microstructure in AM alloys because of the high beam velocity used in AM[45], at the bottom of melt pools where $(G/v_i)$ theoretically becomes infinitely high there should be a change in solidification mode, with planar growth near the bottom of melt pools[19,20]. Examining the fusion lines in this work, it can be seen that, at some fusion lines such as that to the far right of Fig. 1f, the cell array merges into a continuous feature (with thickness being up to 1.0 μm which is significant larger than the cell spacing) that might be due to brief planar front growth and then, after a short growth distance, breaks down again into a cell array. A similar observation of planar growth and subsequent break-down was also seen in high speed electron beam welding of Fe-Cr-Ni alloys[38,39].

**Influence of pores on microstructure development.** 3D printed parts often contain porosity that can lead to changes in the local thermal field and, therefore, alter the solidification microstructure. In AM, porosity can be categorised into three groups based on the

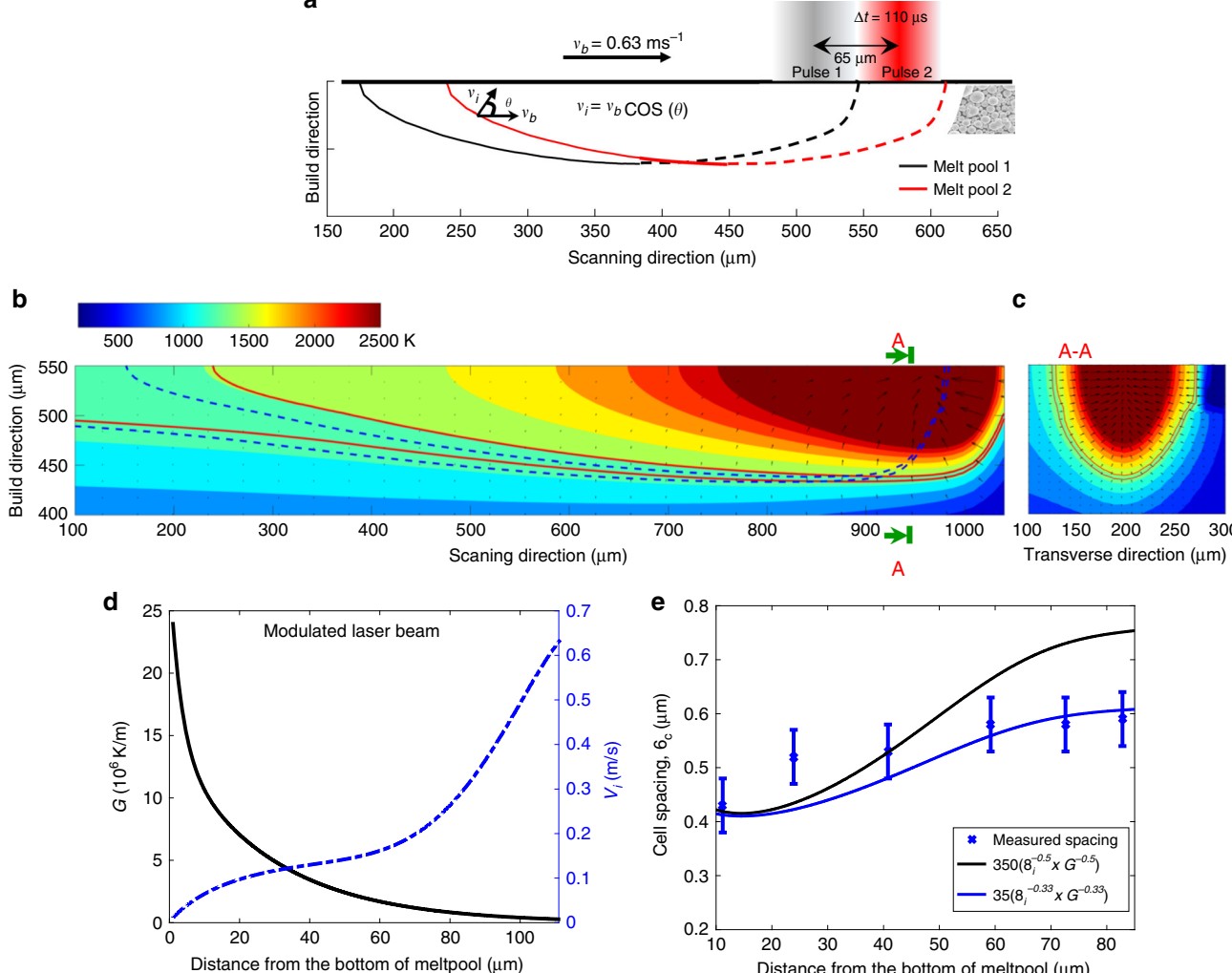

**Fig. 2 Thermal profile in a melt pool of 316L. a** Schematic diagram of two adjacent melt pools (dash lines represent the melting front while the solid lines are for the solidification front); $\mathbf{v_b}$ and $\mathbf{v_i}$ are the velocities of the beam and the solidification front, respectively; $\Delta t$ is the pulsing time of the laser beam. **b** Side-view (parallel to a deposition track) of two consecutive melt pools: the inner and outer solid lines are, respectively, the liquidus and solidus temperatures of the current melt pool while the dashed lines are for the most previous melt pool). The direction and length of vectors represent the direction and magnitude of $G$. **c** A-A section transverse to a deposition track (note: kinks of isotherms on the right of melt pool were due to local changes in heat conduction intentionally introduce to mimic the presence of unmelted powder). **d** The variations of $G$ (solid line) and $v_i$ (dotted dash line) along the liquidus front from the bottom towards the top of a melt pool. **e** Variation of cell spacing: Experimental measurement (error bars represent one standard deviation of uncertainty) of cells shown in Fig. 1d vs. fiting ($v_i^{-m}G^{-n}$) along the distance from the bottom of a melt pool (Note: the identified prefactor of 35 for ($v_i^{-0.33}G^{-0.33}$) was lower than the value frequently used in literature[38, 39, 43, 44] probably because the FEA slightly underestimates the cooling rate. Data are for 316L steel).

mechanism of formation: keyholing, entrapped gases and lack of fusion[5]. Keyholing occurs when excessive power density is used in melting the material, leading to the penetration of the power beam deep into the beneath layers. The collapse of a keyhole results in pores in the bottom of melt pools in 3D printed builds. Although the dimension measurement (Supplementary Fig. 2) of melt pools on the very top layer of 316L fabricated by a Renishaw AM250 indicates the 316L was mainly fabricated in the conduction mode, keyhole pores were frequently observed in the sub-surface of builds (Fig. 3a) probably because of the deceleration and acceleration of beam during turning[10]. Pores formed due to entrapped gases (either due to pre-existing gas inside powder or vaporised material during fusion) are usually spherical and typically smaller than keyhole pores and lack-of-fusion pores (Fig. 3b). Lack of fusion occurs when there is insufficient molten metal to flow to fill gaps (in particular between melt pools), leading to irregular pores (Fig. 3c). The

presence of pores offers an opportunity to observe the 3D morphology of cells. Figure 3b reveals that the solidification microstructure has a layer-like arrangement, where each layer consists of many parallel rod-like cells. The presence of pores can cause disruption to crystal growth depending on the size of pores. While small entrapped gas bubbles did not cause any significant changes in solidification microstructure (Fig. 3b), the large lack-of-fusion pore in Fig. 3c resulted in a striking change in the size and morphology of microstructure. Fine cells exist in regions below the pore while a much coarser microstructure exists in the region right above the pore. The significant role of large pores is because that big pores can serve as thermal insulation, leading to reductions in both $G$ and $v_i$ in the regions above pores. An estimate of the reduction in cooling rate due to the pore in Fig. 3c is about two orders of magnitude (Supplementary Note 2). Interestingly, across the first fusion line away from the lack-of-fusion pore (top region in Fig. 3c),

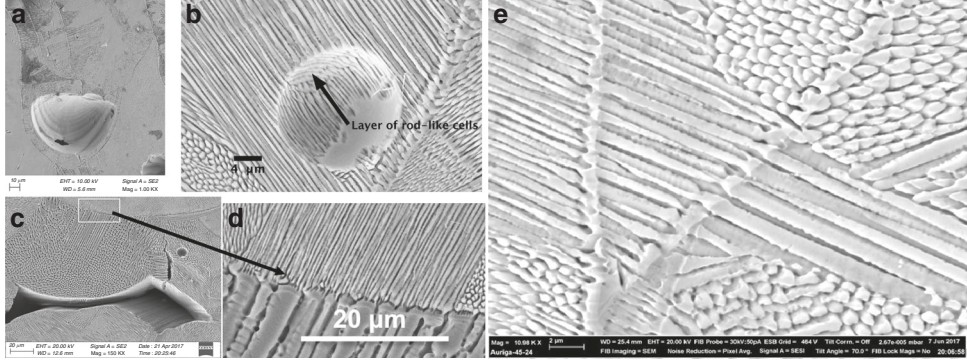

**Fig. 3 Porosity and solidification microstructure in AM 316L. a** Keyhole pore. **b** Layers of fine cells in a region containing a fine spherical entrapped gas pore. **c** Solidification microstructure surrounding a lack-of-fusion pore. **d** Zoom-in region on the centre top of **c** showing a sharp change in cell spacing. **e** Cell refinement occurred twice at two consecutive fusion lines.

coarse cellular dendrites were substantially refined (Fig. 3d). The substantial change in the length scale indicates the cooling rate to be high again once a new melt pool is deposited. Cell refinement was commonly observed at fusion lines between two well consolidated weld beads without the presence of pores (e.g., Fig. 3e). This is similar to the result of cell spacing variation in Fig. 2e, where the bottom of a meltpool had finer cells due to the lowest $v_i^{-0.25}G^{-0.5}$ at the bottom of a meltpool than towards the top of the melt pool.

**Continuous epitaxial growth without changing the direction.** Cells in both the FCC alloys can keep growing across multiple boundaries without changing growth direction in 316L builds fabricated by both modulated and continuous laser exposure strategies (Renishaw—Fig. 4a and Concept Laser—Fig. 4e, respectively) and in HEA builds (Fig. 5d). Such cells were usually confined to the centre of melt pools and being vertical (i.e., parallel to the BD) because **G** along the centreline of melt pool is vertical. The high $G/v_i$ (Supplementary Fig. 3) indicated that columnar growth (without significant nucleation ahead of the growing cell array) is favoured and epitaxial growth is dominant in the evolution of microstructure as frequently reported in literature[8]. Supplementary Fig. 4 shows that if the growth directions of cells in the existing solid are well aligned (within about 30°) to the local **G** at which the cells grew across, the crystals will keep epitaxially growing into the new melt pool without changing their directions.

**Side-branching.** It is often seen in both 316L and HEA that if the growth direction of existing crystals is not preferably aligned to **G**, cells might still be able to epitaxially grow, but with a change in the growth direction (Fig. 4b–d, region 1 in Figs. 4f and 5b). Figure 4b shows three regions of cells (labelled (1), (2) and (3)). Cells in (1) were pre-existing and tangential to the (1,2) fusion line while the cells in (3) were out of plane. The crystallographic orientation map in Fig. 4c shows that the cells in (1) grew along a <001> crystallographic direction (say [100]) inclined to the BD. The cells in (2) grew along a perpendicular <001> direction, say [010]; and the cells in (3) grew along the third perpendicular direction, [001]. For cubic alloys, all three directions are symmetrically equivalent and belong to the <001> family, making cells (1), (2) and (3) belong to the same grain due to epitaxial growth, but have 90° changes in the growth direction. In fact, Fig. 4b, c shows that cells in (2) side-branched out from existing cells in (1). The side-branching is more clearly demonstrated by Fig. 4d which shows that cells in the bottom left melt pool side-branched out from cells in the top right area, causing a change in the growth direction by 90°. Because the local **G** at the fusion line in the new melt pool is perpendicular to the fusion line,

the change of growth direction is driven by the change in the heat flux. Together with the change in local **G**, Fig. 4d highlights the importance of small perturbations on cells as these perturbations offer ready sites for side-branching in response to the heat flux changes, i.e. making side-branching easier to occur. Side-branching often occurred at fusion lines because of the change in **G** once a new melt pool is formed. However, even within a melt pool, **G** significantly varies from location to location. The complexity of the melt pool shape and the dynamics of molten metal can lead to more perturbations in the thermal profile. Cells might initially grow with the preferred direction parallel to the initial local **G**, but as they grow to different regions at which a new local **G** is no longer preferable for the growing cells, promoting side-branching of cells even inside a melt pool (Supplementary Fig. 5a, b). The side-branching from individual cells within a melt pool leads to complex growth paths, e.g. a criss-cross and in-plane structure of cells (Supplementary Fig. 5c), thereby complicated grain morphologies. It is noteworthy that the side-branching within melt pools can cause columnar cells appear to be equiaxed when observed in 2D cross sections via EBSD mapping. Therefore, the interpretation of EBSD mapping of grain microstructure needs to be done with caution.

It should be noted that the continuous growth, tip-splitting and side-branching are seen to be dominant and responsible for microstructure development in both modulated (Fig. 4a–d) and continuous laser systems (Fig. 4e, f), resulting in the statistically same microstructure (Fig. 5a, b and c): e.g slender ([001]//BD)-grains along the centreline of melt pools and ([101]//BD)-grains straddling between two tracks of melt pools (Fig. 5b, c) in the bidirectional scan without rotation. There are only some minor differences: The Renishaw 316L has longer columnar grains (Fig. 5b, c) because of deeper melt pools induced by a higher power intensity (180 W). The same underlying mechanisms seen in the Renishaw and Concept builds are not surprising as the exposure time (60 μs) and spot spacing (60 μm) were very short. Hooper showed that when the beam was on the following spot, the most previous spot was still liquid while the adjacent spot starts to melt[46], making the modulated beam pseudo-continuous. FEA simulation confirms that adjacent melt pools effectively form a pseudo-continuous melt pool over the length scale of about 1 mm though the modulation of the laser beam causes different melt pool profile in transients between melt spots (Supplementary Movies 1 vs. 2). In addition, the underlying mechanisms seen in the 316L steel were also observed in the HEA, e.g. the continuous growth along the centreline and the frequent side-branching on sides of melt pools also result in two sets of thin grains and broad columnar grains in the HEA, respectively (Fig. 5d, e).

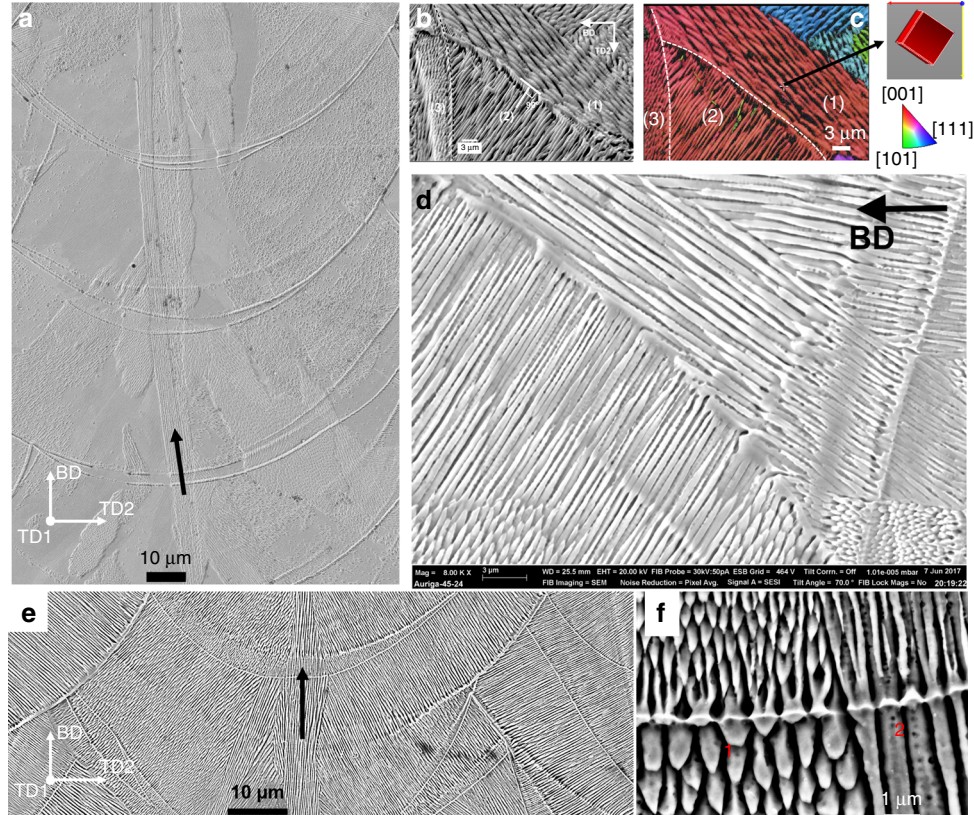

**Fig. 4 The microstructure developments due to continuous growth and side-branching in AM 316L. a** Continuous growth of cells in a slender domain (highlighted by a black arrow) along the centreline across melt pools in the bi-directional scan without rotation. **b–d** Side-branching frequently occurred at sides of melt pools observed in all scan strategies. **b** Cells in (3) epitaxially grew from ones in (2) which did grow from cells in (1), and **c** is a coresponding inverse pole figure along TD1 of region in **a**. **d** Side-branching of cells occurred at a fusion boundary. **e** Continuous growth and **f** sidebranching (region 1) and tip-splitting (regions 1 and 2). Note: 316L steel was fabricated by **a–d** the modulated laser beam (Renishaw) and **e–f** continuous wave laser beam (Concept Laser); the dashed lines in **b** and **c** represent the melt-pool boundary; **b** and **c** were reused with permission provided by AIP Publishing.

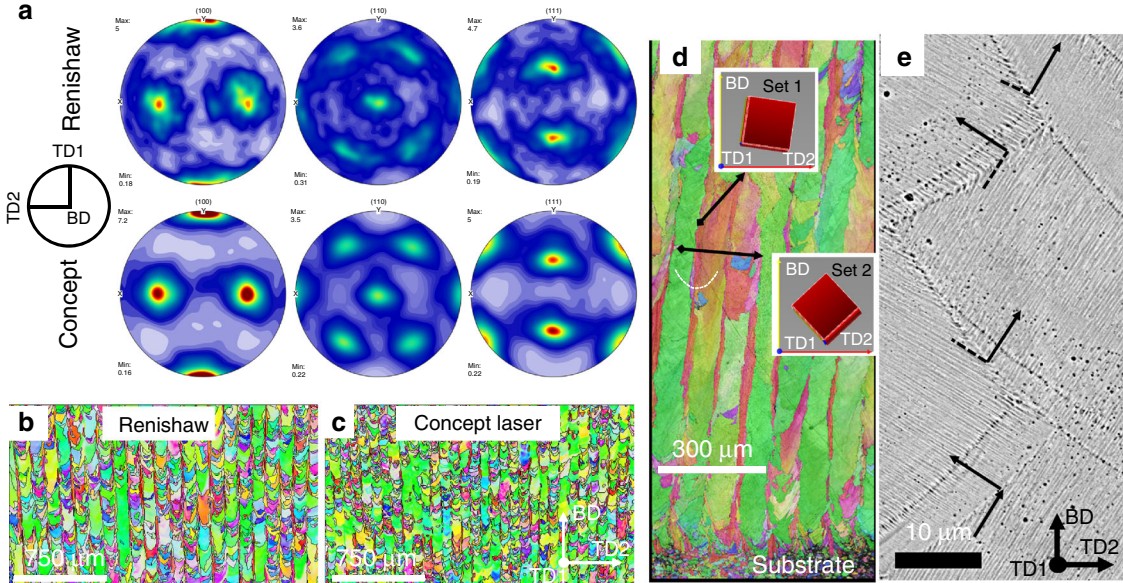

**Fig. 5 Microstructure in the 316L and HEA fabricated by linear bidirectional scanning without rotation for subsequent layers. a** Pole figures of crystallographic orientations observed in 316L fabricated by a Renishaw AM250 and Concept Laser. **b, c** Inverse pole figures along the building direction of crystallographic orientations in 316L fabricated by a Renishaw AM250 and Concept Laser, respectively. **d** Variation of microstructure from the first layer of deposition on a wrought steel substrate towards the top of a build of the HEA fabricated by a Reninshaw AM250. **e** Side-branching dominantly occurred between adjacent melt tracks in a HEA build.

**Roles of scan strategy in microstructure development**. In this section, we use the understanding obtained in the previous sections to demonstrate the influential role of side-branching epitaxial growth in microstructure development in various scan strategies. As all the discussed underlying mechanisms of crystal growth are seen in both 316L and the HEA, this section only discusses microstructure development in the HEA in response to variation in the scan strategy. In scan strategy 1—bidirectional without rotation for subsequent layers—the strategy resulted in predominantly two sets of alternating columnar orientations in the HEA (Fig. 5d, e) as reported in a previous study[13]. One of the two sets is very slender and is confined to the centre of melt pools, with the three <001> directions being well aligned to the BD—build direction, TD1 (i.e., scanning direction) and TD2 which is orthogonal to BD and TD1 (Set 1, Fig. 5). The other set of grains (Set 2, Fig. 5d) had two of the <001> aligned about 45° to BD and TD2 while the third was almost parallel to the TD1. Set 2 was located on the sides of two adjacent melt pools and are much thicker than the first set. Set 1 results from the continuous epitaxial growth across the fusion boundary without changing the direction because cells on the bottom centre of the melt pool tend to be well aligned with **G** along the centreline of melt pools (Figs. 2c and 4a), making a <001> verticle and parallel to the BD. Because of the motion of the laser beam, crystals followed the beam direction; thereby, another <001> is closely parallel to the scanning direction—TD1, making the three <001> well aligned to the sample (BD, TD1, TD2) coordinates. This scan strategy leads to the vertically aligned stacking sequence of melt tracks, promoting cells in the centreline of melt pools to keep growing vertically without changing the cell direction, resulting in the observed thin columnar grains. While cells along the centreline of a melt pool are not affected by the deposition of a newly adjacent melt pool on the same layer, existing cells on the sides of a solidified bead tend to be close to the tangent to adjacent melt pools, promoting side-branching into the adjacent melt pools as highlighted in Fig. 4b–d and Fig. 5e, resulting in the broadening of grains of set 2. Although the grains of set 2 are columnar and vertically oriented, they consist of cells of which a <001> orientation and cell growth direction were not parallel to the BD, but inclined about 45° with respect to the BD due to the alternative side-branching between two adjacent tracks of deposition (Fig. 5e). In addition, it can be seen that similar to the first track (Fig. 1a), the first layers of deposition at the interface between the build and substrate consisted of finer grains (bottom, Fig. 5a). The re-deposition of molten metal leads to the competitive growth of grains[13,21]. Grains consisting of cells that are preferably aligned with the local thermal gradient during melt/re-melt cycles will outgrow the unfavoured ones[21]. It is important to note that the side-branching during repeated deposition allows cells on the sides of solidified beads to easily follow the thermal gradient in new melt pools, substantially broadening grains across two adjacent scan tracks (Fig. 5a, b). In other words, side-branching plays an influential role in the competitive growth. Because grains tried to follow the beam direction, one of the <001> orientations of the set 2 was closely parallel to the scanning direction, similar to that of the first set of orientations.

A scan strategy which rotates the scan pattern between layers alters the alignment of melt tracks and disrupts the thermal profile along the build direction, promoting more random crystal orientations. One of the commonly used strategies to randomise the crystal orientation and reduce the residual stress is the chessboard scan strategy (bottom left inset, Fig. 6a)[26,47]. The chessboard strategy used in this study also significantly changed the grain microstructure. The vertically long columnar grain structure seen in the simple strategy 1 was no longer present (Fig. 5 vs. Fig. 6). Grains were still elongated but substantially shortened. Most grains were inclined to the BD: the top middle inset cube in Fig. 6a shows that two of the <001> orientations of greenish grains were aligned nearly at 45° to the BD, i.e., the grains have the same orientation to the set 2 (Fig. 5a). These grains were able to broaden across not only multiple in-layer melt tracks, but also multiple layers, indicating that side-branching played a significant role. To understand how crystals were able to epitaxially grow across the boundaries between melt tracks of different scan islands, a top view of the thermal profile of a melt track and an EBSD IPF-BD map of a region consisting of two neighbouring islands (I1 and I2) are presented in Fig. 6b, c, respectively. The thermal gradient **G** in a melt pool converges to the hottest region which is the beam location (Fig. 6b). Figure 6b shows that **G** (and thereby grains—Fig. 6c) near the beginning and end of a melt track was well aligned with the centreline. The fast moving beam causes the thermal gradient **G** along the two sides in the rear part of a melt track to be almost perpendicular to the centreline of the melt track (Fig. 6b), making grains grow inwards to and almost perpendicular to the centreline. This means that existing cells on sides of a solidified track are well aligned to **G** in the newly adjacently parallel melt track, providing a preferable condition for continuous epitaxial growth across adjacent melt tracks (e.g., T1, T2 and T3) of the same island. However, in the beginning and end regions of a melt track, **G** is more parallel to the centreline (Fig. 6b), making it easy for existing grains on the sides of a solidified track to epitaxially grow across the island boundaries to newly adjacent, but perpendicular melt tracks that are in a neighbouring island. The 90° change in the scanning direction across the island boundary, therefore, promotes in-layer epitaxial growth between adjacent islands as seen in the region between T1 and T4 (Fig. 6c). Once grains epitaxially grow into a neighbouring island, they can easily further grow from melt-track to melt-track within the same island, making grains penetrate to further inwards islands. Grains in Fig. 6c are rather small because they are of the very top layer (i.e., only undergoing a single melt)—smaller grain microstructure was similar to that on the first layer of deposition on the existing substrate (bottom Fig. 5d); and the subsequent re-melting will broaden grains thanks to side-branching. In addition, the rotation of 67° results in some overlapping of the same islands between consecutive layers (bottom left inset, Fig. 6a). The overlapping promoted the helically out-of-layer side-branching of cells from layer to layer, making cells (thereby, grains) inclined to the BD, e.g., greenish grains in Fig. 6a, top left. It should be noted that EBSD is the 2D mapping of grain orientation, the helically growing grains therefore appear elongated and inclined to the BD as seen in Fig. 6a. Consequently, preferred grains in the chessboard scanning strategy grew helically, leading to an alignment of grain orientations as seen in 2D EBSD mapping of a section perpendicular to the BD (Fig. 6e).

Pole figures after printing with strategy 1 and strategy 2 are presented in Fig. 7a, b, respectively. The (001) pole figure of scanning strategy 1 shows two preferred orientations, reflecting the alternating sets 1 and 2 of orientations shown in Fig. 5d. In contrast to the two dominating sets of orientations, Fig. 7b shows that the <001> orientations associated with the chessboard pattern were angularly rotated around the build direction quite evenly, forming rings along the outmost circle. Equivalently, rings are also seen in the (110) and (111) pole figures with <$\bar{1}$10> well aligned to the BD confirming the presence of a preferred angular

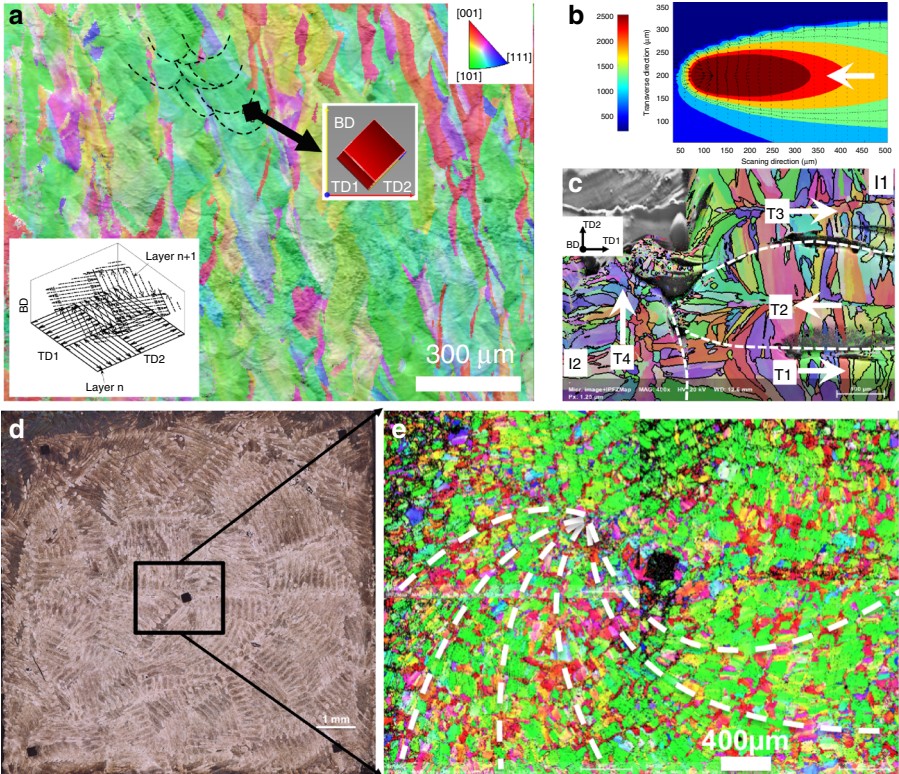

**Fig. 6 Helical growth of grains in the high entropy alloy fabricated by a chessboard scan strategy. a** IPF-BD map of a section along BD of HEA built by the chessboard pattern with a rotation of 67° for every subsequent layer (lower left inset). A big grain of the orientation (shown in the right cube) was able to cross several melt pools in the same layers and across multiple layers (dashed black lines highlight the boundaries of some melt pools). **b** Top-view of the thermal profile in a melt track predicted by FEM simulation; the unit of scale bar is K degree. **c** IPF-BD of a region consisting of two islands I1 and I2, being perpendicular to BD. Arrows in **b** and **c** show the moving direction of the laser beam. White dashed line in **c** indicate the boundaries between melt tracks. **d, e** Optical image and EBSD map (respectively) of a section perpendicular to BD—Note: the section was not perfectly parallel to a single layer of deposition, helping to reveal a spiral microstructure of grains across multiple layers (highlighted by white dashed lines).

texture. This is consistent with the finding of the helically epitaxial growth thanks to in-layer and out-of-layer side-branching shown in Fig. 6, confirming the influential role of side-branching in the development of microstructure when varying the scanning strategy. This influential role also explains a strong cube texture for the bidirectional linear scanning with the 90° rotation between layers. The cube texture has all three <001> very well aligned to BD, TD1 and TD2 (Fig. 7c) in agreement with previous studies for cubic crystals[48,49]. The 90° rotation of scanning direction in every layer leads the deposition to alternatingly out-of-layer side-branch along TD1 and TD2, making two <001> orientations aligned to TD1 and TD2, resulting the observed cube texture.

The most influential effect of preferred crystallographic texture is seen on the plastic anisotropic behaviour[50] that is identified as one of the main concerns for additive manufacturing[6,12]. The measured hardness clearly reflects the influential effect of varying the scan strategy on the plastic anisotropy. The bi-directional scanning with 90° (strategy 3) exhibits most isotropic behaviour: measured hardness on Z-sections was similar to that on X-section (Fig. 7d). The alignment of <001> orientations with the (BD, TD1 and TD2) means that cubic crystals should behave the same in BD, TD1 and TD2, resulting in the most observed isotropy in agreement with previous studies[50,51]. In contrast, different epitaxial growths in the strategies (1) and (2) lead to distinctive crystallographic texture and strong anisotropy: hardness on Z-sections was substantially lower than that on X-sections (Fig. 7d). Taylor factors measured on the basis of texture shown in Fig. 7a–c

using MTEX for the loading direction parallel to BD (Z section) and perpendicular to BD (X section) were 3.20 (and 3.61) and 3.75 (and 3.83) for strategy 1 (and 2), respectively. Higher Taylor factors usually result in higher macroscopically measured stresses[50], explaining why the hardness measured on Z section was lower than that on X section for the two strategies 2 (Fig. 7d).

## Discussion

Microstructure development in powder-bed fusion of cubic alloys has been studied in relation to the local thermal gradient—**G** and isotherm velocity—$v_i$ to explain the change in microstructure under different scan strategies. The present study shows that crystal growth without changing direction often occurs and is confined to the centreline of melt pools, resulting in long columnar but slender domains. The misalignment between existing crystal cells and **G** promotes side-branching from the perturbations on the sides of existing cells onto a perpendicular <001>, leading to epitaxial growth with a change in the cell growth direction. The role of side-branching is influential as it results in a 'criss-cross' layer microstructure and broadening of grains in the subsequent deposition in 3D printed alloys. In particular, side-branching is responsible for microstructure development when varying the scanning strategy. Most interestingly, the chessboard strategy with 67° rotation between layers breaks the vertical columnar grain microstructure, but it promotes both in-layer epitaxial growth and out-of-layer side-branching, resulting in helical epitaxial growth. It has been shown

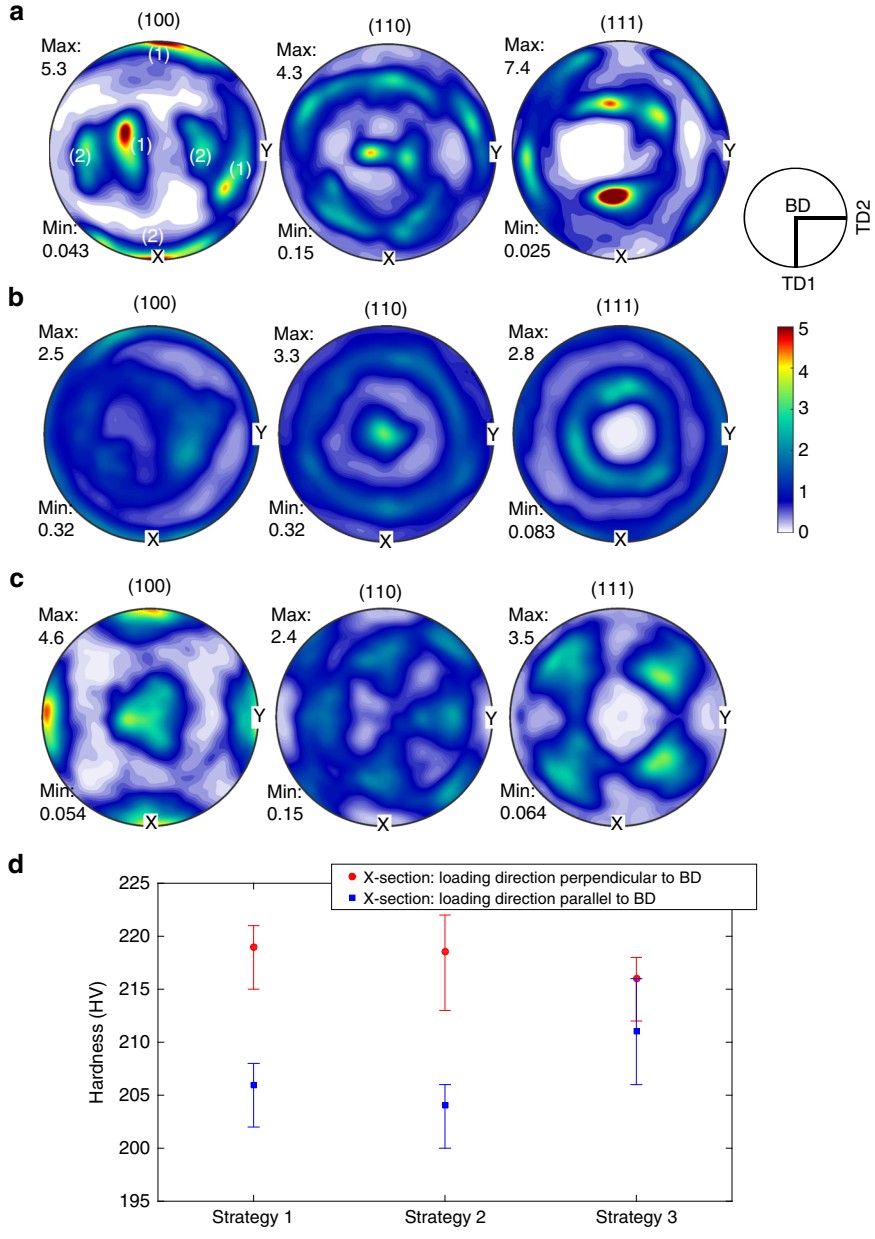

**Fig. 7 Crystallographic orientations and plastic anisotropy associated with three different scan strategies.** Pole figures of HEA samples fabricated by **a** strategy 1: bi-directional scanning without rotation, **b** strategy 2: chessboard with 67° rotation, and **c** strategy 3: bi-directional scanning with 90° rotation (Note: X is parallel to TD1, Y is parallel to TD2 and BD is at the centre of the pole figures). **d** Hardness measurements when loading direction being parallel and perpendicular to BD. (The upper and lower limits of the error bars represent the 25th and 75th percentile of the measurements, respectively).

that variations in the length-scale of microstructure correlates well with $v_i^{-0.25}G^{-0.5}$, and large pores cause a substantial coarsening of the microstructure due to their local thermal insulating effect.

## Methods

**Materials and experiments.** This study examines the microstructure development in cubic crystal phases that are the most frequently reported phase to be formed during the solidification of austenitic stainless steels, Ni, Al and Ti alloys. Thermal cycles associated with the repetitive deposition in AM promote the solid state phase transformation in Ti alloys or precipitation-hardened alloys such as Ni and Al alloys. The precipitation and presence of multiple phases would make the interpretation of observed microstructure difficult. To avoid complications associated with the solid phase transformation, this study was only focused on single phase alloys. Two cubic alloys, stainless steel 316L and high entropy alloy CrMnFeCoNi, were selected because they are both single-phase face-centred cubic (FCC) and similar solidification behaviour and printability. In addition, the examination of

microstructure in the two alloys help to confirm the validity of underlying mechanisms (continuous growth without direction change, side-branching) discussed in this study.

Austenitic stainless steel 316L (provided by LPW Technology Ltd) and a high entropy alloy (HEA) CrMnFeCoNi (purchased from H.C. Starck Surface Technology & Ceramic Powders GmbH) were studied. The composition of the HEA and 316L steel powders is shown in Supplementary Table 1. The size distributions of powder of the two alloys are given in Supplementary Fig. 6.

Two types of experiment were performed by laser powder bed fusion (LPBF) using a Renishaw AM250 printer. First, in order to understand how the microstructure forms in rapid cooling, an eleven-track single-layer build of the HEA was printed with a uni-directional scan strategy on a 316L stainless steel substrate. A wide hatch spacing of 125 μm was used in the single-layer build to minimise individual tracks from experiencing the thermal cycles originating from the deposition of adjacent tracks.

In the second type of experiment, multi-layer builds were printed to provide samples for subsequent examination of crystal growth. A bi-directional hatch pattern (laser beam linearly moves back and forth) was used to melt powder onto existing solidified layers. This study focuses on linear bi-directional scans for the

whole area of deposition with the rotation of 0° and 90° for subsequent layers; and a chessboard strategy with rotation of 67°. In the chessboard scan strategy, a layer of material deposition is divided into square domains (i.e., islands) similar to the squares of a chessboard. Every island is deposited by a linear bi-directional scan pattern, but the pattern is rotated by 90° for its in-layer adjacent neighbours. The bi-directional scan with the rotation of 67° was also used to give a better overview of microstructure. The HEA was printed as cubes of $10 \times 10 \times 10$ mm with a power intensity of 200 W, an exposure time of 80 μs and a point distance of 60 μm (equating to a linear scan speed of approximately 0.75 m s$^{-1}$) while 316L multi-layer builds were printed with various sizes and following parameters: laser power of 180 W; exposure time of 110 μs; point distance of 65 μm (giving a linear scan speed of approximately 0.6 m s$^{-1}$). The hatch spacing (125 μm), layer height (50 μm) and laser spot size (65 μm) were kept constant for both materials. An argon atmosphere was used for both alloys to protect the molten metal from oxidation.

The modulated laser beam in Renishaw might affect the epitaxial growth. To confirm that the validity of the underlying mechanisms observed in builds fabricated by modulated beam is also applicable to materials fabricated by continuous wave laser, 316L samples were also printed by Concept Laser with following print parameters: bi-directional scan strategy without rotation, power of 90 W, scan speed of 600 mm/s, spot size of 50 μm, hatch spacing of 77 μm and layer thickness of 20 μm.

The samples were sectioned along the build direction (BD) and perpendicular to the BD using an alumina blade on a Struers Accutom-50. After mechanical polishing by silicon carbide particle papers, they were further polished by diamond suspension containing fine particles of different sizes (6, 3 and 1 μm). These specimens were finally polished by a vibratory polisher with colloidal silica solution consisting of 0.05 μm particles mixed with 7 vol% of $H_2O_2$. To reveal the solidification microstructures, polished samples were electro-chemically etched (at 5 V and 2.5 V for 316L and HEA, respectively) for 90 s in an electrolyte solution of 10% oxalic acid in water. Microstructural features such as grain size and morphology, crystallographic orientation and crystal phases were studied by field emission scanning electron microscopes (FEG-SEMs): Zeiss™ Sigma 300 and Zeiss™ Auriga Cross Beam. The latter is equipped with a high resolution Bruker e-FlashHR electron backscattered diffraction (EBSD) detector. EBSD maps were subsequently analysed by Bruker Esprit 2.0 software. X-ray diffraction was done in a Bruker D2 PHASER with the 2θ from 9- 99° (angle increment of 0.036° and time per step of 0.5 s). Hardness measurements were carried out by applying a 2 kg load for 10 s on sections perpendicular to the BD (i.e., applied load was parallel to the BD) and on sections parallel to the BD (i.e., applied load was parallel to the BD). For each section at least 15 indentations were made in a regular array.

**Simulation of the thermal field**. The microstructure in LPBF are strongly dependent on the thermal profile in the moving melt pool. Whilst it is possible to measure surface temperature profiles directly[46], there is no method to measure temperature gradients from within the melt pool experimentally and simulation tools must be used to predict these values. A finite element (FE) model based on simplified LPBF process was therefore developed. The purpose of this model was not to accurately capture all the physical phenomena of melt pool dynamics and laser material interaction, but to reasonably estimate the melt pool shape, size and thermal gradients at the solid-liquid interface in a single track and to aid in the interpretation of the observed microstructures.

A model was created using the commercial FE package Abaqus. The laser energy input was represented as a moving volumetric heat source using the Goldack double ellipsoidal heat source model[52]. A laser power of 180 W (with assumed coupling efficiency of 0.6[53–55]). The coupling efficiency was estimated by interpolating the powder absorptivity for power of 180 W and scan speed of 0.63 m s$^{-1}$ experimentally measured by Trapp et al.[55]. Both the modulated and continuous laser modes were set up to have a velocity of 0.63 m s$^{-1}$ and a bi-directional scan path with hatch spacing of 125 μm was used. The volumetric heat input $q$ at time $t$ and moving along $x$ direction with the velocity of 0.63 m s$^{-1}$ is described as a function of position $x$, $y$ and $z$ as follows[52,56]:

$$q(x,y,z,t) = \frac{6\sqrt{3}Qf_f}{a_f\pi\sqrt{\pi}(bc)}e^{-\left(\frac{x-x_0-0.63t}{a_f}\right)^2}e^{-\left(\frac{y-y_0}{b}\right)^2}e^{-\left(\frac{z-z_0}{c}\right)^2}$$
$$\mp \frac{6\sqrt{3}Qf_r}{a_r\pi\sqrt{\pi}(bc)}e^{-\left(\frac{x-x_0-0.63t}{a_r}\right)^2}e^{-\left(\frac{y-y_0}{b}\right)^2}e^{-\left(\frac{z-z_0}{c}\right)^2}-q_i$$

(2)

with $x_0$, $y_0$, $z_0$ and $x$, $y$, $z$ are the centre location of the laser beam and the location where the heat flux was calculated $a_f$, $a_r$, $b$ and $c$ are semi-axes of an ellipsoid with centre located at $(x_0, y_0, z_0)$. $f_f$ and $f_r$ are the fractions of the deposited heat from the front and rear quadrants of the source respectively defined as described in[57] $q_i$ is the necessary heat input to heat up the deposition bead from the substrate temperature (25 °C for the first track of deposition) to a nominal melting temperature, 1442 °C. $q_i$ was set to be 1100 kW m$^{-3}$[56]. $Q$ is the energy input rate.

The semiaxes of the Goldack heat source model were calibrated so that the melt pool size matched that experimentally measured to be: 90 ± 20 μm in depth and 145 ± 30 μm in width for melt pools on the very top layer of 316L fabricated by Renishaw (Supplementary Fig. 2). The values for $a_f$, $a_r$, $b$ and $c$ were identified to be

35 μm, 135 μm, 120 μm and 35 μm respectively. Isotropic heat transfer can be described as follow[58]

$$\rho\left(\frac{\partial H}{\partial t}\right) = \nabla(k\nabla T)$$

(3)

with following boundary conditions Preheat temperature: $T = T_0 = 25$ °C, Input heat flux: $(-k\nabla T) \cdot \hat{n} = q$, Surface convection: $(-k\nabla T).\hat{n} = h(T - T_0)$, where $\rho$ is the density, $H$ is the enthalpy, $k$ is the thermal conductivity (value of $k$ provided in[56] was used in this study), $h$ is the film convection coefficient (5.7 W m$^{-2}$ K$^{-1}$ and assumed to be temperature independent), and $\hat{n}$ is the normal vector to the surface of the domain.

We used a Renishaw AM250 with a very short exposure time (110 μs) and short spot spacing (60 μm), making a collective continuous "melt pool" with length being up to about 1.2 mm which is across about 19 nominal melt spots. In addition, the validation shows that the model provides accurate prediction of melt pool dimensions and cooling rates that were experimentally measured (Supplementary Note 1). A modelling domain of 1.5 mm long × 0.5 mm wide and 0.7 mm high was used with a 316L stainless steel 50 μm powder layer at the top and a solid layer beneath. Linear 8-node brick elements of type DC3D8 (diffusive heat transfer variation) were used for their simplicity and low computational cost. The mesh size in the powder layer was 10 μm × 10 μm × 5 μm with increasing size in the lower sections of the model for computational efficiency. The powder layer was modelled with temperature dependent material properties with low thermal conductivity and density[54,58,59]. A dynamic material model was used to allow the transformation from powder to bulk material to be captured. This is achieved using a user-defined 'USDFLD' subroutine in Abaqus. Upon reaching a temperature of 1442 °C the material model properties change from those for the powder to those for the consolidated bulk material. The density for powder was found to be about 58 to 60% that of the consolidated material[59]. In this study, the densities for powder was 4699.9 kg m$^{-3}$ about 59% that of the consolidated material (7966.0 kg m$^{-3}$). The liquid phase of the material is not explicitly modelled but its thermal properties are captured through the use of temperature dependent material properties in the consolidated material model[56]. Latent heat of fusion (with a specific latent heat of 275 kJ kg$^{-1}$[53]) was also included in the material model and was applied in the temperature range between liquidus and solidus (1442 °C and 1325 °C, respectively). It is expected that he fluid affects the thermal profile in the melt pool. It is worth to note that Knapp et al.[60] shows the inclusion of fluid in FEA simulation leads to a marginal difference (about 10% in the cooling rate) towards the terminal stage of solidification (i.e. near the top of melt pool which is often remelted in multi-layer deposition). Therefore, the fluid was not included in the simulation to reduce the computational cost. Liquidus and solidus temperatures were obtained via Scheil-Gulliver model simulation for the 316L steel in using Thermo-Calc Software TCFe database version 7. This allows to account for the change in solidification range during rapid cooling. In the lower section of the model the heat capacity was increased to account for the large thermal mass of the base plate. A film convection coefficient of 5.7 W m$^{-2}$ K$^{-1}$ was also applied to the top surface[61]. This provides some cooling effect, but it is almost negligible compared to conductive cooling into the solid material. The calculated values of solidus and liquidus temperatures were used to identify the solidus and liquidus isotherm interfaces in the thermal profile obtained by FEA. Supplementary Tables 2, 3 summarise the values of parameters used for FEA simulation in this study.

The results obtained from the thermal field simulation were then analysed in Matlab. The temperature data from nodes on sections of interest were extracted and fitted using the 'griddata' function. Thermal gradients and solidification velocities were calculated using the interpolated gridded data at the liquidus isotherm. The data was fitted with 7$^{th}$ degree polynomial for plotting. The R$^2$ of the fit were >0.997.

## Data availability

The datasets generated during and/or analysed during the current study are available from the corresponding author on reasonable request.

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

## Acknowledgements

The authors thank Mohanad Bahshwan for providing 316L samples fabricated by Concept Laser. This work was supported by the Imperial College London's Engineering Alloys fellowship awarded to MSP. MSP thanks financial support provided by EPSRC (Grant EP/K503733/1) and a Feasibility studies fund sponsored by EPSRC-MAPP grant (EP/P006566/1).

## Author contributions

M.S.P. directed the research, performed the electron microscopic examination and the analysis of the crystal growth. M.S.P. wrote the manuscript with significant contributions from all the other authors. B.D. carried out chemical polishing, electron backscattered diffraction pattern, hardness measurements and FEA simulation. P.A.H. fabricated the 316L and HEA by a Renishaw and set up FEA simulations. C.M.G. contributed to the analysis of thermal profile and solidification. A.P. performed electron backscattered diffraction pattern and electron microscopic imaging.

## Competing interests

The authors declare no competing interests.
