## [Peer Review File · Nature Communications]

Reviewers' Comments:

Reviewer #1:

Remarks to the Author:

The authors produced a full length paper on the development of the solidification cell structure due to laser powder bed fusion. The findings are interesting and important for the understanding of the microstructural development in this process. The reviewer notes the following points for the authors to consider:

1. The authors presented the results for both 316L and HEA. It would have been sufficient to focus on a single alloy, as the inclusion of both alloys only increased the length of the manuscript, rather than identified clear differences in the way both alloys behave with regards to solidification. The conclusions fail to highlight any obvious differences between both alloys. The rationale behind including the two alloys need to be explained. The work on 316L in itself is sufficient, and the HEA work should go into a future paper.

2. The work was conducted on Renishaw platform, which is known to have a start-stop quasi pulsed/modulated laser mode, which differs from the majority of powder bed platforms (Concept GE, EOS, SLM Solutions) which use a continuous wave laser. Some of the correlations that the authors performed with other published work were with builds on other platforms. This needs to be highlighted throughout. I suggest using Todd et al. Acta Mat. paper on nominal energy density to correlate the energy density levels used in this study with other platforms. Also, I am not sure if the used scanning strategy correlated with the 67° rotation on the EOS platform. The same applies for the chess/island scanning strategies in Concept GE and SLM Solutions platforms. These difference are essential to be highlighted in the work.

3. Please clarify the face on which the XRD measurements were performed. In most XRD studies conducted for AM builds that use continuous wave laser for cubic materials (e.g. using Concept GE or EOS), the cube 001 direction reflections become stronger, unlike what is shown in Fig. 3.

4. The thermal model involves many assumptions as it was not thermally validated. I presume the start-stop Renishaw operation mode was not considered. Can the authors correlate the cooling rates (Fig. 2-d) with the cell size (Fig. 5)? The reviewer suggests that the authors correlate the cell size measurements with papers from the literature (look up a thesis by John Walter Elmer, LLNL).

Otherwise, the reviewer finds the findings worth publishing in nature communication.

Reviewer #2:

Remarks to the Author:

This reviewer agrees with authors that significant science challenges need to be overcome for metal additive manufacturing (AM) to be a key enabler in the 4th industrial revolution. The manuscript is well written. It shows a very good effort to make high quality contribution to further the understanding of microstructure formed during solidification under the highly complex and dynamic condition of laser powder bed fusion (LPBF) AM. This includes the simulation work and calculation to obtain thermal gradient and liquidus isotherm and relating them to cell (dendritic) growth, and how scan strategy may be controlled to vary texture developed during LPBF and subsequently to plastic anisotropy.

There are however number of questions and comments authors should consider:

1. The size of LPBF samples has not been clearly provided and modelling calculation has been on very small area. Modelling results (G and v) have been used to relate to measured cell spacing values. Then, some discussion/explanation of sample/modelling sizes may need to be made for the

validity of relating the thermal condition modelled to the measured cell spacing.

2. More information of both LPBF condition and XRD sample should be provided for Fig. 3. The XRD patterns are provided to show FCC phase. But there should be a very strong preferred orientation in 90 degree and to a less extent in 67 degree as scanned samples. Then, since XRD patterns are given, should not they show preferred orientation?

3. In III.1.3 (then later in III.1.4), in regard to $m=n=0.5$, have most used $m=n=0.33$? In ref 27, the reference authors have cited, Qui et al. use 0.33.

4. The cross section provided in Fig. 4 for both single track and multi-track/layer samples have not evidence of keyhole mode dominant. What presented in Fig. 6a (and not in other micrographs) is suggested to have resulted from keyhole mode. What mode(s) operated during the present LPBF experiments then need to be clarified and explained.

5. In III.2.1 Continuous growth (and after), regarding "crystal orientation [001] perpendicular to the boundary": in the pool melt boundary, the growth direction (the moving direction of liquidus isotherm, locally) during solidification is perpendicular to the melt boundary, fusion line, but [001] is not exactly the same as the growth direction. The difference, meaning the angle between [001] and the growth direction can be small but can also be quite high. Authors should make clear of this. For example, in Fig. 8, cells in region 1 is perpendicular to cells in region 2 and these two regions are separated by fusion line as authors have stated. But cells in region 1 are not necessary and have not been determined to be normal to the fusion line and cells in region 2 are not necessary and have not been determined to be parallel to fusion line. Cells (or primary arms) are straight but fusion line are not unless in scan direction.

6. (Referring to Fig. 8, and also an important point in the manuscript), authors state "the importance of small perturbation on cells as these perturbations offer ready sites for side-branching, leading to the 90 degrees change in growth direction in response to the heat flux changes". Does this then suggest pure cellular growth cannot result in 90 degrees change in growth direction even the heat flux should cause the change? This does not seem reasonable.

7. For the non-long-columnar grains in 90 degrees (or 0 degree) scan rotation sample, grain growth with $\langle 001 \rangle$ aligned to 45 degrees to vertical direction is shown in Fig. 10b. But for 67 degrees scan rotation, "most grains grains with $\langle 001 \rangle$ crystallographic orientation aligned ~ 45 degrees to the BD" is not seen convincing.

8. Referring to Fig. 11, authors state small grains in top layer and remelting would lead to large grains. This may be unclear and need to be better demonstrated and explained.

A few minor points:

1. Has Fig. 4c mentioned in text?
2. In the paragraph after Fig. 6, the last sentence, should it be Fig. 6c, not 10c?
3. In the sentence just before Fig. 9, one of the two "Fig. 9c"s can be deleted.
4. The 10th line under III.3., should it be Fig. 10a, instead of Fig. 10b?
5. Fig. 5e left, the top and bottom map do not seem too well lined up.

Reviewer #3:

Remarks to the Author:

The manuscript describes laser powder bed additive manufacturing of SS316 and a CrMnFeCoNi high-entropy alloy and seeks to characterize and rationalize the development of the grain structure relative to the melt pool heat transfer conditions. The paper presents interesting characterization of the material structure which may be of interest to the broader AM community. However, the

level of analysis of the underlying mechanisms is lacking, and this work contributes little new scientific understanding to microstructure development in AM. Unfortunately, this research is ultimately not suitable for publication in Nature Communications. Please see below for detailed comments.

1. The authors imply that the primary difference between cellular and dendritic growth is the presence of secondary arms. This is not the case. Cellular growth is characterized by a lack of a strong dependence of the growth direction on the crystallography (see ref. 10) and cellular growth would, by definition, not exhibit most of the characteristics shown in this work.
2. The influence of dendrite side-branching has been characterized by the welding community and well understood for nearly 30 years, most notably in a classic work by Rappaz et al. (Met. Trans. A, 1990). The present work does little to expand on this understanding, and notably, does not reference the seminal research in this area.
3. Similarly, epitaxial growth of grains through multiple layers of an AM build has been described in detail in a large variety of papers, including studies on the influence of scan pattern, and is not a novel finding.
4. The numerical modeling is not presented in sufficient detail. The governing equations, material properties, and boundary conditions are not quantified. The heat source model was calibrated against experimental data, but the comparison with experiments is not shown, and the calibrated values are not reported. The mesh size seems large compared to similar computational work done in this area, but a mesh independence study is not reported. The methodology for calculating the thermal gradients and solid-liquid interface velocities is not reported.
5. The transient behavior of the solidification conditions is a primary concern in this work. However, the authors do not consider the pulsed nature of the Renishaw AM250 laser, and instead, simply approximate it as a continuous beam. With a pulsed beam, the solidification conditions may be oscillatory, with significant variation in the local solidification conditions. However, the assumption of a continuous beam is made without supporting evidence.
6. The paper is much too long, and greatly exceeds the recommended word limit for this journal. Given the technical content, I do not think this length is warranted. The writing is generally unfocused.
7. Data in Figures 4a and 10b are taken directly from ref 7 with no indication of permission from the original publisher.

Responses to Reviewers' comments

Reviewer #1 (Remarks to the Author):

The authors produced a full length paper on the development of the solidification cell structure due to laser powder bed fusion. The findings are interesting and important for the understanding of the microstructural development in this process. The reviewer notes the following points for the authors to consider:

1. The authors presented the results for both 316L and HEA. It would have been sufficient to focus on a single alloy, as the inclusion of both alloys only increased the length of the manuscript, rather than identified clear differences in the way both alloys behave with regards to solidification. The conclusions fail to highlight any obvious differences between both alloys. The rationale behind including the two alloys need to be explained. The work on 316L in itself is sufficient, and the HEA work should go into a future paper.

R1:

We thank the Reviewer for the compliment and the suggestions. We wanted to include both the alloys to reflect the broad validity of the study. In this revised manuscript, we provided additional data (Fig. 5, Supplementary info, Fig. S1 and Notes S3) and added text in Section II.3 to highlight the similarities and differences between two alloys. We also emphasize the need of studying multiple alloys (Methods, Materials) and demonstrate the similarity of microstructure development phenomena (continuous growth – (Fig. 4a and Fig. 5a), side-branching – Fig. 4b-d and Fig. 5b and Supplementary info-Notes S3) in 316L and HEA to confirm that the same observed underlying mechanisms are responsible for microstructure development in the two alloys.

2. The work was conducted on Renishaw platform, which is known to have a start-stop quasi pulsed/modulated laser mode, which differs from the majority of powder bed platforms (Concept GE, EOS, SLM Solutions) which use a continuous wave laser. Some of the correlations that the authors performed with other published work were with builds on other platforms. This needs to be highlighted throughout. I suggest using Todd et al. Acta Mat. paper on nominal energy density to correlate the energy density levels used in this study with other platforms. Also, I am not sure if the used scanning strategy correlated with the 67° rotation on the EOS platform. The same applies for the chess/island scanning strategies in Concept GE and SLM Solutions platforms. These difference are essential to be highlighted in the work.

R2:

The authors wanted to thank you the Reviewer for this important point. Following the Reviewer's comment, we examined microstructure in 316L fabricated by Concept Laser Mlab that uses a continuous laser exposure. As reported in the revision (Section II, Supplementary info – Notes S3), we found the underlying mechanisms (continuous and side branching epitaxial growth) are dominant and responsible for the similar microstructure development in builds fabricated by both continuous laser exposure (Concept Laser) and modulated laser exposure (Renishaw) despite of different in energy densities. The different energy densities lead to some detailed differences such as the cell spacing, grain size and the intensity of preferred texture.

3. Please clarify the face on which the XRD measurements were performed. In most XRD studies conducted for AM builds that use continuous wave laser for cubic materials (e.g. using Concept GE or EOS), the cube 001 direction reflections become stronger, unlike what is shown in Fig. 3.

R3: We provided more detail information regarding the sections on which X-ray diffraction was done (Supplementary info, Fig. S1 - Caption). It was found that while HEA has strong (200) peak, 316L has (200) weaker than (111) and (220). We discussed this difference (Supplementary info). We found that our measured XRD profile of 316L is in agreement with a previous study (ref. 29) done by a group in USA on 316L fabricated by Concept Laser.

4. The thermal model involves many assumptions as it was not thermally validated. I presume the start-stop Renishaw operation mode was not considered. Can the authors correlate the cooling rates (Fig. 2-d) with the cell size (Fig. 5)? The reviewer suggests that the authors correlate the cell size measurements with papers from the literature (look up a thesis by John Walter Elmer, LLNL).

Otherwise, the reviewer finds the findings worth publishing in nature communication.

R4:

The Reviewer was correct that we did not account for the pulse of laser beam. Please refer to the response R2 regarding the study of microstructure in pulsed vs continuous wave beam. We thank the reviewer to pointing out a very interesting and important study that helps us to discuss our study in greater depth. Following the suggestion, we used relationship between cell size and

cooling rate ($\lambda_c = av_i^{-0.33}G^{-0.33}$) reported in Elmer's studies (refs. 39, 40) and other previous papers (refs. 44, 45) to experimentally estimate the cooling rate to be in the range of 0.5×10^6 to 1.6×10^6 K/s (Supplementary info-Notes S1). We re-ran our simulation with much finer mesh size and presented detailed description of FEA modelling (Methods, Simulation of the thermal field, pages 20-22). We also provided the model validation (Supplementary info-Notes S1) based on the measurements of dimensions of melt pool and cooling rate. The matching of the melt pool dimensions and cooling rates between experimental measurements and simulation confirmed that the FEA provided reasonably accurate prediction.

Please refer to the response **R19** below concerning the effect of pulse beam vs continuous beam.

Reviewer #2 (Remarks to the Author):

This reviewer agrees with authors that significant science challenges need to be overcome for metal additive manufacturing (AM) to be a key enabler in the 4th industrial revolution. The manuscript is well written. It shows a very good effort to make high quality contribution to further the understanding of microstructure formed during solidification under the highly complex and dynamic condition of laser powder bed fusion (LPBF) AM. This includes the simulation work and calculation to obtain thermal gradient and liquidus isotherm and relating them to cell (dendritic) growth, and how scan strategy may be controlled to vary texture developed during LPBF and subsequently to plastic anisotropy.

There are however number of questions and comments authors should consider:

1. The size of LPBF samples has not been clearly provided and modelling calculation has been on very small area. Modelling results (G and v) have been used to relate to measured cell spacing values. Then, some discussion/explanation of sample/modelling sizes may need to be made for the validity of relating the thermal condition modelled to the measured cell spacing.

R5: We provided the details of sizes of fabricated builds in the Methods (pages 19-20) and simulation (pages 20-22). We added text (page 20) to clarify our

simulation aim that was to estimate the thermal profile in single track to assist our interpretation of cellular growth in relation to thermal profile in single tracks. We presented the validation of simulation on the basis of comparing the simulated melt pool size against the experimentally measurement and the simulated cooling rate against the calculated cooling rate on the basis of measured cell spacing (Supplementary info-Notes S1). The calculated G , v and measured cell spacing were used in discussion presented in Section II, pages 5 and 6.

2. More information of both LPBF condition and XRD sample should be provided for Fig. 3. The XRD patterns are provided to show FCC phase. But there should be a very strong preferred orientation in 90 degree and to a less extent in 67 degree as scanned samples. Then, since XRD patterns are given, should not they show preferred orientation?

R6: the details of the samples used for XRD were provided in Supplementary information Fig S1. The purpose of providing XRD data was to confirm that the two alloys are single phase. We provided additional explanations regarding the texture observed in XRD (refer to our response **R3**). It was found that while HEA has strong (200) peak, 316L has (200) weaker than (111) and (220). We discussed about this difference (Supplementary info). We found that our measured XRD profile of 316L is in agreement with a previous study (ref. 29) done by a group in USA on 316L fabricated by Concept Laser.

3. In III.1.3 (then later in III.1.4), in regard to $m=n=0.5$, have most used $m=n=0.33$? In ref 27, the reference authors have cited, Qui et al. use 0.33.

R7:

We thank the Reviewer the suggestion. We used different values of m and n (including $m=n=0.33$, i.e. $\lambda_c = av_i^{-0.33}G^{-0.33}$) to fit the variation of cell size in the revised manuscript – page 6 (also refer to the response **R4**).

4. The cross section provided in Fig. 4 for both single track and multi-track/layer samples have not evidence of keyhole mode dominant. What presented in Fig. 6a (and not in other micrographs) is suggested to have resulted from keyhole mode. What mode(s) operated during the present LPBF experiments then need to be clarified and explained.

R8: We added discussions regarding the measurement of melt pool dimensions (depth versus width) indicates the fabrication was mainly done in the conduction mode, but in the transition between the conduction and keyhole modes as keyhole pores could be found in subsurfaces at which material was

often over heated during the deceleration and acceleration of beam during turning (page 8).

5. In III.2.1 Continuous growth (and after), regarding “crystal orientation [001] perpendicular to the boundary”: in the pool melt boundary, the growth direction (the moving direction of liquidus isotherm, locally) during solidification is perpendicular to the melt boundary, fusion line, but [001] is not exactly the same as the growth direction. The difference, meaning the angle between [001] and the growth direction can be small but can also be quite high. Authors should make clear of this. For example, in Fig. 8, cells in region 1 is perpendicular to cells in region 2 and these two regions are separated by fusion line as authors have stated. But cells in region 1 are not necessary and have not been determined to be normal to the fusion line and cells in region 2 are not necessary and have not been determined to be parallel to fusion line. Cells (or primary arms) are straight but fusion line are not unless in scan direction.

R9: We revised the corresponding text (last sentence of Section II.1 - first paragraph, and throughout the manuscript) to specific that *crystal orientation [001]* and the growth direction were closely normal to the fusion boundary at which the cells grew across. Text corresponding to Fig. 4 (formerly Fig8) in the first paragraph (Section II.3.2) was also revised to clearly state that cells in Region 1 were pre-existing (and tangential) to the melt pool boundary of Region 2.

6. (Referring to Fig. 8, and also an important point in the manuscript), authors state “the importance of small perturbation on cells as these perturbations offer ready sites for side-branching, leading to the 90 degrees change in growth direction in response to the heat flux changes”. Does this then suggest pure cellular growth cannot result in 90 degrees change in growth direction even the heat flux should cause the change? This does not seem reasonable.

R10: We are sorry for any unclear statement. The sentence meant that the perturbations promote the side-branching (i.e., making the side-branching easier to happen). This does not mean to exclude the side-branching of pure cells. We edited the last sentence of the 1st paragraph of Section II.3.2 to make it clearer.

7. For the non-long-columnar grains in 90 degrees (or 0 degree) scan rotation sample, grain growth with <001> aligned to 45 degrees to vertical direction is

shown in Fig. 10b. But for 67 degrees scan rotation, “most grains grains with <001> crystallographic orientation aligned ~ 45 degrees to the BD” is not seen convincing.

R11: We revised the text in the lines 4, 5 and 6 (page 15) to make our statement clearer in particular we specifically referred to the middle top inset cube that shows two of the three <001> crystallographic orientations of grains to be aligned about 45 degrees to the BD (note that Fig. 10 in the previous version was now Fig. 6 in the revised version).

8. Referring to Fig. 11, authors state small grains in top layer and remelting would lead to large grains. This may be unclear and need to be better demonstrated and explained.

R12:

We provided additional data (Fig 5a) clearly showing the evolution of grain microstructure from fine grains in the first layer to coarser grains toward the top of a multi-layer build to demonstrate the competitive growth leads to the coarsening of grains. We found the side-branching allows cells to more easily follow the new heat flux and broaden during repeated deposition. Text was added (page 12) to describe the competitive growth and the role of side-branching in the growth. We also added a sentence “Smaller grain microstructure ...” highlighted on page 15 to describe the small grain microstructure on the very top layer is similar to that of the first layer of deposition. Note that Fig. 11 is now Fig. 6 in the revised version.

A few minor points:

- 1. Has Fig. 4c mentioned in text?*
- 2. In the paragraph after Fig. 6, the last sentence, should it be Fig. 6c, not 10c?*
- 3. In the sentence just before Fig. 9, one of the two “Fig. 9c”s can be deleted.*
- 4. The 10th line under III.3., should it be Fig. 10a, instead of Fig. 10b?*
- 5. Fig. 5e left, the top and bottom map do not seem too well lined up.*

R13:

1. Fig. 4c (now Fig. 1c) was referred in text (page 3).
2. This paragraph is now moved the supplementary information and the reference to the figure was corrected. Note: Fig. 6c is now Fig. 3c.
3. Fig. 9 was moved to Supplementary info (now Figure S3). One mention of Fig.3c was removed.
4. Sub-figures of Fig. 10 were replaced by new images, and Fig. 10 was renumbered to be Fig. 5. The wrong cross reference was corrected.
5. We were very sorry that there was no Fig. 5e in the first version of

manuscript. We would be very grateful if the reviewer could clarify the suggestion.

Reviewer #3 (Remarks to the Author):

The manuscript describes laser powder bed additive manufacturing of SS316 and a CrMnFeCoNi high-entropy alloy and seeks to characterize and rationalize the development of the grain structure relative to the melt pool heat transfer conditions. The paper presents interesting characterization of the material structure which may be of interest to the broader AM community. However, the level of analysis of the underlying mechanisms is lacking, and this work contributes little new scientific understanding to microstructure development in AM. Unfortunately, this research is ultimately not suitable for publication in Nature Communications. Please see below for detailed comments.

R14:

We greatly appreciated the Reviewer's comment. Although there are plenty of previous studies reporting the epitaxial growth is the main mechanism responsible for the microstructure development in additive manufacturing, none of these clearly show how the microstructure spatially develops. In fact, one of our previous studies was the first report showing a columnar grain microstructure in cubic alloys consists of two alternating orientations that locate at specific locations and have distinctive morphologies (ref 13). In addition, all previous studies (including our own) fail to reveal the underlying phenomena that result in different orientations and morphologies of columnar grains at different locations. Moreover, no previous studies highlight the important role of side-branching in enabling crystals to follow the change of heat flux in response to the variation of deposition strategy in AM, i.e., side-branching is influential in the competitive growth, broadening and spatial orientation of grains in AM. AM is known for its powerful capabilities in tailoring microstructure to specific location. Clear and detail insights into the governing role of side-branching epitaxial growth are important and significant as they will help AM users to develop better ways and increase the confidence in tailoring microstructure to specific locations, thereby properties. We made substantial revisions to the manuscript (in particular the introduction) to clearly highlight the significance of the present study. To the authors' knowledge, is the first study revealing how continuous growth and side-branching are responsible for different columnar microstructures at different locations and the roles of side-branching in the competitive growth and broadening of microstructure, offering significant insights into understanding and explaining the microstructure development in AM.

1. The authors imply that the primary difference between cellular and dendritic growth is the presence of secondary arms. This is not the case. Cellular growth is characterized by a lack of a strong dependence of the growth direction on the crystallography (see ref. 10) and cellular growth would, by definition, not exhibit most of the characteristics shown in this work.

R15:

We revised the first paragraph (page 5). In particular, we referred to seminal studies by Mullins & Sekerka and Langer (refs 31, 32). According to refs 31 and 32, the microstructure pattern is governed by the instability of liquid-solid interfaces. The dendritic pattern results from the instability of cell/dendrite tip. Thereby, the presence of secondary dendrites is often seen to be indication of dendritic microstructure. In fact, multiple experimental and computational studies done by Trivedi, Karma, Georgelin (ref. 33-37) differentiated cells from dendrites based on the tip radius and liquid/solid front instabilities (that leads to formation of secondary arms). In particular, Trivedi *et. al.* measured spacings of cells and dendrites, and the critical spacing (above which cells transform to dendrites) on basis of the morphology of microstructure (features were called dendrites when there were clear secondary arms – referred to Fig. 2 in ref 36). Ref. 36 also shows the formation of secondary arms in the transition from cells to dendrites (Fig. 9 in ref 36). Georgelin & Pocheau (ref. 34) and Seetharaman *et. al.* (ref. 37) showed that cells also grew along one of $\langle 001 \rangle$ directions. In addition, Georgelin & Pocheau also highlighted that while the orientation of all cells in the same domain is homogeneous (i.e., the preferred $\langle 001 \rangle$ is the same for every cell in the domain). These characteristics were also seen in our study of cells. Following these studies, we adopt the differentiation of cells and dendrites based on the presence of secondary arms. It should note that a study of Rappaz *et. al.* (ref. 22 as suggested by the Reviewer) also named microstructure observed in laser welding as dendritic cells (see the abstract of ref. 22). Similarly, some other studies (co-authored by Rappaz and Kurz) also described microstructure in laser weld of FCC alloys the transition between cells and dendrites (eg, ref 38 - Wang *et. al.*, *Acta Mat.*, 52, 11, p 3173-3182; ref. 20 - Gäumann *et. al.*, *MSE A271* (1999), 232-241). In particular, in ref. 20, Gäumann *et. al.* also differentiated the difference between cells and dendrites based on the presence of side arms.

*2. The influence of dendrite side-branching has been characterized by the welding community and well understood for nearly 30 years, most notably in a classic work by Rappaz *et al.* (*Met. Trans. A*, 1990). The present work does little to expand on this understanding, and notably, does not reference the seminal*

research in this area.

R16:

We thanked the Reviewer for referring to this seminal study and pointing out our omission. We discuss Rappaz's study in the revised manuscript (ref. 22). In particular, we wanted to highlight that the Rappaz's study focused on the crystal growth in single-pass welding (with laser speed of tens of mm/s) on a solid substrate of a single crystal. This study, indeed, provides many insights in to crystal orientations in the weld bead for different scenarios of the relationship between the orientation of the existing single crystal and the beam direction. However, the situation in laser powder-bed fusion is quite different and more complex because the substrate in AM usually consists of many crystal grains of different orientations. The AM process also involves many passages of deposition with the beam speed of hundreds to thousands of mm/s, and the beam direction varies from melt track to melt track and from layer to layer. Our present study clearly shows how the microstructure develops in multiple deposition on polygrain substrates under various deposition directions with scan speed of about 600 mm/s; thereby, it is distinctive itself from Rappaz's study. Most importantly, reference 22 did not study the significant role of side-branching that was found in our study to be the most important to the development of microstructure in AM. To reflect this distinction, we made a major revision to the introduction (pages 2 and 3)

3. Similarly, epitaxial growth of grains through multiple layers of an AM build has been described in detail in a large variety of papers, including studies on the influence of scan pattern, and is not a novel finding.

R17:

We agree with the Reviewer that there are plenty of studies stating that epitaxial growth through multiple layers of deposition is responsible for the microstructure in AM builds. The most comprehensive review of epitaxial growth in AM is presented in ref 8 showing that most of previous studies reported that columnar grains with [001] orientation // build direction is commonly observed in AM builds. However, none of previous studies clearly unravel how epitaxial growth is responsible for the spatial detail of morphologies and orientations (of microstructure) at different locations in melt pools and how the microstructure characteristics develop spatially with the variation in scan strategies (also refer to the response **R16**). In this study, we presented significant insights to explain the development of columnar grains at different locations in melt pools: slender ones with [001] orientation // build direction in regions along the centreline of melt pools, and broadened columnar grains of [101] // BD across adjacent melt pools. We also reported

new findings that explain the observed microstructure; and to our knowledge, first time showing the roles of side-branching in competitive growth, broadening and spatial development of grains under the variation in scan strategies. We believe our study presents significant new findings that will increase our confidence in controlling microstructure in specific locations, thereby properties; consequently helps to solve materials-related issues that are identified as a major problem in AM.

4. The numerical modeling is not presented in sufficient detail. The governing equations, material properties, and boundary conditions are not quantified. The heat source model was calibrated against experimental data, but the comparison with experiments is not shown, and the calibrated values are not reported. The mesh size seems large compared to similar computational work done in this area, but a mesh independence study is not reported. The methodology for calculating the thermal gradients and solid-liquid interface velocities is not reported.

R18:

We thanked the Reviewer for this comment. In the revision, we provided details of modelling including governing equations (Eqn. 2 and 3, page 21) describing the heat input and heat transfer. Boundary conditions were also presented in Pages 20, 21 and 22. We re-ran the simulation with much finer element size (8 times). Details of calculating the liquidus and solidus interfaces were also given (Page 22). Simulation was validated by comparing the melt pool dimensions and cooling rates between the experimental measurement and simulation. The description of the validation is provided in the Supplementary info Note S1 (please refer to the response **R4** for some further details).

5. The transient behavior of the solidification conditions is a primary concern in this work. However, the authors do not considered the pulsed nature of the Renishaw AM250 laser, and instead, simply approximate it as a continuous beam. With a pulsed beam, the solidification conditions may be oscillatory, with significant variation in the local solidification conditions. However, the assumption of a continuous beam is made without supporting evidence.

R19:

We revised our manuscript to highlight the focuses of our study which are to understand the underlying mechanisms (continuous growth and side-branching) responsible for the development of microstructure multi-deposition with various scan strategies in AM. We found that that show that the

continuous growth and side-branching are applicable to both modulated and continuous wave beam (Supplementary info, Notes S3), resulting in similar microstructure observed in 316L fabricated by both a Renishaw AM250 (modulated beam) and a Concept Laser (Continuous beam). The same underlying mechanisms seen in Renishaw and Concept builds are not surprising as we used Renishaw with a very short exposure time (60 μ s) and short spot spacing (60 μ m). Hooper showed that when the beam was on the following spot, the most previous spot was still liquid (ref 47 - Hooper, Additive Manufacturing 22, 548-559, 2018), making a continuous “melt pool” with length being up to 1 mm which is across more than 10 nominal spots, i.e. it is reasonable to assume the modulated beam pseudo-continuous. Therefore, it is reasonable to assume that the energy source was continuous as we did on the simulation. In addition, as mentioned in Responses R4 and R18, we validated the simulation (by comparing the dimensions of melt pools and cooling rates – Supplementary info Notes S1) to further confirm that the FEA provided reasonably accurate prediction. We reflect this justification by adding text in the beginning of 2nd paragraph, page 22.

We did not observe obvious effects (such as evidence of oscillatory solidification) of pulsing of the beam on microstructure. In fact, Fig. S2 as well as data presented in the main text show that the epitaxial growth is very influential in the microstructure development in both the modulated and continuous beam. Correspondingly, we added text throughout Section III.3 and III.4 to reflect the applicability of the reported underlying mechanisms for modulated and continuous laser systems.

6. The paper is much too long, and greatly exceeds the recommended word limit for this journal. Given the technical content, I do not think this length is warranted. The writing is generally unfocused.

R20:

We have revised the presentation of the study throughout the manuscript, in particular we shortened the manuscript and made it much more focused within the word limit. We made substantial edits to highlight the mechanisms responsible for the microstructure development. We moved the X-ray diffraction profile, the calculation of the variation in cooling rates due to the presence of pores, validation of simulation and the measurement of angles between cells and fusion boundary to the Supplementary info. We have provided substantially additional information and corresponding discussion (Section II.3)) to demonstrate that mechanisms reported in this study are responsible for microstructure in modulated vs continuous beam

(Supplementary info, Notes S3). We also restructured the manuscript by arranging the Materials, Experimental and Simulation sections into Methods.

7. Data in Figures 4a and 10b are taken directly from ref 7 with no indication of permission from the original publisher.

R21:

We added notes where the permission of reuse images is required, e.g. “Note: Figure (a) was re-used from [13] under the terms of the Creative Commons Attribution License (CC BY)” for the re-use of Fig. 4a (now 1a). Figure 10b was removed and replaced by a different image (Fig. 5a) in the revision.

Reviewers' Comments:

Reviewer #1:

None

Reviewer #2:

Remarks to the Author:

I have reviewed having checked point by point response and the revised manuscript. Authors have addressed the points satisfactorily. Two small points: (1) in line 129, it should be Fig. 2e? (2) Referring to R13, point 5, in my original comment (minor point) I made a mistake and it should be Fig. 11e in the originally submitted version, not Fig. 5e. In the revised version it is Fig. 6e. The comment is still the same for the revised version and the top right map probably needs to be shifted up slightly. Authors please check.

Reviewer #1 had four comments. My view on Authors' responses:

R1: In reviewer #1's opinion, it is not necessary to present the work for two alloys in the paper as "the work on 316L in itself is sufficient". But it appears it is only reviewer 1's view, and reviewer #1 has not sounded it mandatory to change. Authors has not revised buying changing to only presenting the work on 316L. Instead, authors have explained the need to include the another alloy so that findings are more applicable to other alloys.

R2: Reviewer #1 points to pulsing of Renishaw machine which is different to other powder bed platforms. Authors has replied that the side-branching phenomenon also occurs using other platforms. Authors also present this in the revised version. Thus, the underlying mechanisms, which is primarily the focus of this paper is the same. As the paper does not deal qualitatively with the frequency of side-branching, I view authors have addressed satisfactorily.

R3. Reviewer #1 points out "the cube 001 direction reflections become stronger, unlike what is shown in Fig. 3". Authors has revised by providing more information and point out the agreement of their data with others. It is possible, in my view, authors have not quite explained what reviewer 1 would like to see the explanation.

R4. Authors have done in revision and satisfied in regard to reviewer #1's question "can the authors correlate the cooling rates with the cell size".

Reviewer #3:

Remarks to the Author:

Thank you to the authors for a detailed response and the effort put forth for revision. I have several outstanding comments that should be addressed before publication.

1. I still do not agree with the usage of the term "cells" here. I understand that this has become common usage in the AM and welding literature for describing dendrites without secondary arms. However, this does not change the fact that the behavior observed here is more accurately described as dendritic. In any case, the authors clearly observe instabilities indicative of the early stages of secondary arm formation.
2. The explanation for the change in morphology at the melt pool boundary shown in Figure 1f is that it is caused by low velocity planar solidification. Could there be an effect of partial remelting of material within the alloy mushy zone?
3. I'm not sure I follow the discussion of the values of the constants for the primary dendrite spacing. Based on these results, $m=0.5$ and $n=0.25$ is deemed appropriate. But the values of the

gradient and velocity are derived from a model with a large number of simplifying assumptions. The selection of these constants therefore seems arbitrary. You also use $m=n=0.33$ in validating the cooling rate of your model. So is the model validated? Or did you use the model to determine the appropriate constants? Both cannot be true.

4. While the data in Figure s4 suggests that similar mechanisms for grain selection are active for both systems, I'm not sure that it supports the modeling of the pulsed laser as an effective continuous energy source. I would much prefer a numerical comparison of the distributions of the thermal gradients and velocities at the solid-liquid interface under these assumptions which would leave no doubt as to the validity of this assumption.

5. I find Figure s4 to be extremely informative, and more so than the figures in the main manuscript. The pole figures clearly show the bulk texture develop that results from the interaction of the growth directions relating to the crystal symmetric and the selected scan pattern. I recommend that this figure and associated discussion be moved to the main manuscript. If there is not room, I would suggest removing some of the HEA results. I agree with Reviewer #1 that the value of reporting the HEA results is not clear. I do not find the argument that the HEA reinforces the "broad validity of the study" particularly compelling. It is well known that cubic crystal structures preferentially select $\langle 001 \rangle$ type growth directions, which is the dominant feature of those materials that explains your observations.

6. In Figure 6, if the cross-sections shown in (d) and (e) are perpendicular to the build direction, why are multiple scan rotations observed? Is there some sectioning artifact that reveals the spiral structure? Within a horizontal section, the scan rotation should be uniform, correct?

7. I appreciate the detail in the description of the numerical model, but have a few additional questions. Why was fluid flow neglected? Is this a good assumption? Recent research (Knapp et al., Additive Manufacturing, 2019) shows fluid flow is important in the trend in solidification conditions. How was the energy absorption efficiency of the 0.6 selected? This is quite high, particularly considering that radiation and vaporization are neglected. How were the effective properties of the powder calculated? Also, I would prefer a table with a complete list of material properties, perhaps in the supplementary material. Why do you say that the overall thermal profile obtained from SS316 simulations is applicable to the HEA? How different are the thermophysical properties? If the model results are so general as to be simultaneously applicable to different alloys, then I'm not sure the results can be used quantitatively.

Responses to Reviewers' comments

Reviewer #1 had four comments. My view on Authors' responses:

R1: In reviewer #1's opinion, it is not necessary to present the work for two alloys in the paper as "the work on 316L in itself is sufficient". But it appears it is only reviewer 1's view, and reviewer #1 has not sounded it mandatory to change. Authors has not revised buying changing to only presenting the work on 316L. Instead, authors have explained the need to include the another alloy so that findings are more applicable to other alloys.

R2: Reviewer #1 points to pulsing of Renishaw machine which is different to other powder bed platforms. Authors has replied that the side-branching phenomenon also occurs using other platforms. Authors also present this in the revised version. Thus, the underlying mechanisms, which is primarily the focus of this paper is the same. As the paper does not deal qualitatively with the frequency of side-branching, I view authors have addressed satisfactorily.

R3. Reviewer #1 points out "the cube 001 direction reflections become stronger, unlike what is shown in Fig. 3". Authors has revised by providing more information and point out the agreement of their data with others. It is possible, in my view, authors have not quite explained what reviewer 1 would like to see the explanation.

R4. Authors have done in revision and satisfied in regard to reviewer #1's question "can the authors correlate the cooling rates with the cell size".

Response 1:

We thank the feedback on our response to the Reviewer 1's comment R1 - 4. Although the Reviewer approved our responses to the comments R1, 2 and 4, we revised the text to increase the coherency of the study (last sentence on Page 12 and 1st paragraph in Section II.4)). We added text in the methods (page 20) to better reflect our justification of the choice of the two alloys: confirming the validity of the observed underlying mechanisms in more than one FCC alloys. We provided the simulation of pulsed mode. Please refer to our response 6.

In response to the comment R3, we provided additional text below the figure S1 in the supplementary info to explain why we did not see a strong (001) peak in the diffraction profile of the 316L.

Reviewer #2:

I have reviewed having checked point by point response and the revised manuscript. Authors have addressed the points satisfactorily. Two small points: (1) in line 129, it should be Fig. 2e? (2) Referring to R13, point 5, in my original comment (minor point) I made a mistake and it should be Fig. 11e in the originally submitted version, not Fig. 5e. In the revised version it is Fig. 6e. The comment is still the same for the revised version and the top right map probably needs to be shifted up slightly. Authors please check.

Response 2

We thank you the Reviewer for the comment and clarification!
We corrected the typos and re-aligned the top right of Figure 6 to make them more aligned to the bottom right.

Reviewer #3:

Thank you to the authors for a detailed response and the effort put forth for revision. I have several outstanding comments that should be addressed before publication.

1. I still do not agree with the usage of the term “cells” here. I understand that this has become common usage in the AM and welding literature for describing dendrites without secondary arms. However, this does not change the fact that the behavior observed here is more accurately described as dendritic. In any case, the authors clearly observe instabilities indicative of the early stages of secondary arm formation.

Response 3:

We greatly appreciated the insightful and critical comments offered by the Reviewer. As we described in the manuscript (highlighted text, page 5), the microstructure is in the transition from cells to dendrites. Therefore, we agree that describing the observed microstructure as dendrites is also correct. Nevertheless, as most of solidification features have weak perturbations on sides, indicating the transition is closer to the cell development. Therefore, we would stay with the use “cells” and be consistent with the use in literature as cited in the manuscript.

2. The explanation for the change in morphology at the melt pool boundary shown in Figure 1f is that it is caused by low velocity planar solidification. Could there be an effect of partial remelting of material within the alloy mushy zone?

Response 4:

We agree that the remelting of existing cells also can generate layers connecting the existing cells. However, if there was only cellular growth, cells would grow right from from these bridge layers, making the layer very thin (for example, Figure 4d (top right – near the arrow) and Fig. 4f). The measurement of the layer thickness shown in Fig. 1f was about 0.5 μm to 1.0 μm which is almost equal and larger than the cell spacing. The significantly thick layer in Fig. 1f suggests that the solidification mode was briefly planar. We reflected this discussion by adding text on Page 6.

3. I'm not sure I follow the discussion of the values of the constants for the primary dendrite spacing. Based on these results, $m=0.5$ and $n=0.25$ is deemed appropriate. But the values of the gradient and velocity are derived from a model with a large number of simplifying assumptions. The selection of these constants therefore seems arbitrary. You also use $m=n=0.33$ in validating the cooling rate of your model. So is the model validated? Or did you use the model to determine the appropriate constants? Both cannot be true.

Response 5:

We validated the simulation on basis of the melt pool dimensions – Supplementary info Notes S1. This method is typically used to validate the FEA simulation in welding and AM, including the study by Knapp referred by the Reviewer.

As presented in the Notes S1, we also used a $\lambda_c = 80v_i^{-0.33}G^{-0.33}$ which is found in literature to be well represent the relationship between the spacing of cells and the cooling rate (refs 39, 40, 44 and 45 in the Main text – or 7, 8, 9 and 10 in the Supplemental info) to further confirm the prediction of cooling rate. We revise the text in the simulation validation, Suppl. info Notes S1.

In addition, we checked that the simulated solidification front velocity was consistent with the used beam velocity, and compared the thermal gradient (predicted by our FEA simulation) against those reported in literature (Supplemental info, page 5).

4. While the data in Figure s4 suggests that similar mechanisms for grain

selection are active for both systems, I'm not sure that it supports the modeling of the pulsed laser as an effective continuous energy source. I would much prefer a numerical comparison of the distributions of the thermal gradients and velocities at the solid-liquid interface under these assumptions which would leave no doubt as to the validity of this assumption.

Response 6:

We conducted simulation and provided simulation results for modulated and continuous beams (Fig. 2 – modulated, Fig. S3 – continuous; Videos S1 – modulated and S2 - continuous). The results show that the thermal profile when a spot is fully melted in the modulated beam (Fig. 2) is similar to the thermal profile in the continuous beam (Fig. S3), and multiple melt spots join together to make a big collective melt pool which is similar to a melt pool in the continuous beam, though the shape of collective melt pools is different to that in the continuous melt pool due to transients between melt spots (Videos S1 and 2).

5. I find Figure s4 to be extremely informative, and more so than the figures in the main manuscript. The pole figures clearly show the bulk texture develop that results from the interaction of the growth directions relating to the crystal symmetric and the selected scan pattern. I recommend that this figure and associated discussion be moved to the main manuscript. If there is not room, I would suggest removing some of the HEA results. I agree with Reviewer #1 that the value of reporting the HEA results is not clear. I do not find the argument that the HEA reinforces the “broad validity of the study” particularly compelling. It is well known that cubic crystal structures preferentially select <001> type growth directions, which is the dominant feature of those materials that explains your observations.

Response 7:

We thank the reviewer for the suggestion. In response to the suggestion, we now use the Figure S4 in the main text. Fig. S4a-d were now Fig. 5a-c, and Fig. S4e-f were now Fig. 4e-f. The associated text of Fig. S4 was now incorporated in the main text, Pages 11 and 12.

We revised the text to increase the coherency of the study (last sentence on Page 12 and 1st paragraph in Section II.4)). We added text in the methods (page 20) to better reflect our justification of the choice of the two alloys: confirming the validity of the observed underlying mechanisms in more than one FCC alloys.

6. In Figure 6, if the cross-sections shown in (d) and (e) are perpendicular to the

build direction, why are multiple scan rotations observed? Is there some sectioning artifact that reveals the spiral structure? Within a horizontal section, the scan rotation should be uniform, correct?

Response 8:

The reviewer was correct that the cross section (Fig. 6d and e) was not absolutely perpendicular to the BD. It was a little bit off from 90 to the BD due to the limitation in of mechanical cutting and polishing. This limitation provides advantages in helping to reveal the helical growth: the cross section was slightly off from 90 to the BD means that the cut was through some multiple layers, enabling the observation of crystal growth in the same island in consecutive layers, making it easier to see how cells grow across multiple layers. A note was accordingly added in Fig 6.

7. I appreciate the detail in the description of the numerical model, but have a few additional questions. Why was fluid flow neglected? Is this a good assumption? Recent research (Knapp et al., Additive Manufacturing, 2019) shows fluid flow is important in the trend in solidification conditions. How was the energy absorption efficiency of the 0.6 selected? This is quite high, particularly considering that radiation and vaporization are neglected. How were the effective properties of the powder calculated? Also, I would prefer a table with a complete list of material properties, perhaps in the supplementary material. Why do you say that the overall thermal profile obtained from SS316 simulations is applicable to the HEA? How different are the thermophysical properties? If the model results are so general as to be simultaneously applicable to different alloys, then I'm not sure the results can be used quantitatively.

Response 9:

We thank the Reviewer for raising this point and the reference co-authored by Knapp and colleagues. We agree that the fluid can affect the thermal profile in the melt pool as it can change the heat transfer in the melt pool. It is worth to note that Knapp and colleagues' study (ref. 61) shows the effect of fluid is rather marginal (e.g., the inclusion of fluid in FEA simulation led to 10% shorter cooling time compared to the no inclusion of fluid). The study also shows that most of effect associated with the consideration of fluid is seen at the very terminal stage of cooling (i.e. at the very top centre of a melt pool). In multi-layer deposition, top of melt pool is usually re-melted (for example, in our study, the melt pool depth was $90 \pm 20 \mu\text{m}$ while the layer thickness was $50 \mu\text{m}$, i.e. about half on the top of melt pool was remelted). Microstructure change due to the inclusion of fluid will be removed by the redeposition of material in the next layer. Therefore, we did not include the fluid in our

simulation. We reflected this discussion in the manuscript (Pages 23, 24).

The value of 0.6 for the energy absorption coefficient was obtained from fitting the reported measurement for 316L powder (ref 56) and interpolating for the print parameters of 180W and scan speed 0.63 m/s used in our study. Text was added to reflect this justification (Page 22).

Spierings *et al.* (ref. 60) The density for powder was found to be about 58% to 60% that of the consolidated material. Therefore, we identified the density of powder to be about 59% that of the consolidated material. Regarding the thermal conductivity, we followed an assumption made in a study authored by Hussein *et al.* (ref 55) (in which the thermal conductivity coefficient of powder was about 10% of the bulk at temperatures below solidus temperature). Supplemental info Table 2 summarises the values of thermal conductivity and specific heat capacity.

In response to the Reviewer's comment, we now provided two tables (including relevant references) that summarize the values of parameters used in our simulation (Supplemental info Tables 2 and 3 - Pages 2, 3).

We agreed that the data obtained for 316L could not be quantitatively applied to the HEA (indeed we did not use the simulation data of 316L for the HEA in this study). We rephrased text on page 5 to clarify our statement that the understanding of the thermal profile and solidification behaviour can be applicable to the HEA as the two alloys shared underlying solidification mechanisms.

Reviewers' Comments:

Reviewer #3:

Remarks to the Author:

Thank you for the thorough revision. I am satisfied that the manuscript is suitable for publication.